# Pretraining Generative Flow Networks with Inexpensive Rewards for Molecular Graph Generation

**Mohit Pandey** [1]   **Gopeshh Subbaraj** [2]   **Artem Cherkasov** [1]   **Martin Ester** [3]
**Emmanuel Bengio** [4]

## Abstract

Generative Flow Networks (GFlowNets) have recently emerged as a suitable framework for generating diverse and high-quality molecular structures by learning from rewards treated as unnormalized distributions. Previous works in this framework often restrict exploration by using predefined molecular fragments as building blocks, limiting the chemical space that can be accessed. In this work, we introduce Atomic GFlowNets (A-GFNs), a foundational generative model leveraging individual atoms as building blocks to explore drug-like chemical space more comprehensively. We propose an unsupervised pre-training approach using drug-like molecule datasets, which teaches A-GFNs about inexpensive yet informative molecular descriptors such as drug-likeliness, topological polar surface area, and synthetic accessibility scores. These properties serve as proxy rewards, guiding A-GFNs towards regions of chemical space that exhibit desirable pharmacological properties. We further implement a goal-conditioned finetuning process, which adapts A-GFNs to optimize for specific target properties. In this work, we pretrain A-GFN on a subset of ZINC dataset, and by employing robust evaluation metrics we show the effectiveness of our approach when compared to other relevant baseline methods for a wide range of drug design tasks. The code is accessible at https://github.com/diamondspark/AGFN.

[1]Vancouver Prostate Centre, The University of British Columbia [2]Mila - Quebec AI Institute, Université de Montréal [3]Simon Fraser University [4]Recursion Pharmaceuticals . Correspondence to: Mohit Pandey <mkpandey@student.ubc.ca>, Gopeshh Subbaraj <gopeshh.subbaraj@mila.quebec>.

*Proceedings of the $42^{nd}$ International Conference on Machine Learning*, Vancouver, Canada. PMLR 267, 2025. Copyright 2025 by the author(s).

## 1. Introduction

GFlowNets are amortized samplers that learn stochastic policies to sequentially generate compositional objects from a given unnormalized distribution, i.e. a reward signal. They can generate diverse sets of high-reward objects, which has a demonstrated utility in small molecules drug-discovery tasks (Bengio et al., 2021). Traditionally, GFlowNets used for molecule generation have focused on fragment-based drug discovery, i.e. their action space comprises of some predetermined fragments as building blocks for molecules. While producing diverse candidates, fragment-based approaches limit the pockets of accessible chemical space (Zhu et al., 2024; Bengio et al., 2021; Jain et al., 2023; Shen et al., 2024). A truly explorative generative policy, tapping into the full potential of GFlowNet would be realizable when using atoms instead of fragments as the action space (Mouchlis et al., 2021; Meyers et al., 2021; Nishibata & Itai, 1991). On the other hand, the vastness of this accessible state space makes training atom-based GFlowNets susceptible to collapse. Earlier atom-based GFlowNets attempt to overcome this issue by limiting models to small trajectories (Jain et al., 2023). However, since most commercially available drugs have molecular weights ranging from 200 to 600 daltons(Bos & Meinardi, 2000; Bickerton et al., 2012), the molecules generated from these small trajectories are unlikely to possess the drug-like characteristics necessary for therapeutic efficacy. In this work, we propose to scale the trajectory length of atom-based GFNs by pretraining them with expert demonstrations, coming in the form of offline trajectories generated from a dataset of drug-like molecules. Our main contributions in this paper are as follow:

- We introduce A-GFN, an atom-based GFlowNet for sampling molecules in proportion to rewards governed by inexpensive molecular properties.

- We propose a novel strategy for unsupervised pretraining of A-GFNs by leveraging drug-like molecules using offline, off-policy training. This pretraining enables broader exploration of chemical space while maintaining diversity and novelty relative to existing drug-like molecule datasets.

- We demonstrate that goal-conditioned finetuning of A-GFNs for sampling molecules with desired properties offers significant computational advantages over training A-GFNs from scratch.

- Finally, we demonstrate the practical utility of A-GFN finetuning for optimization of structual and pharmacological properties of drugs, *de novo* generation of high-affinity binders, and for lead optimization.

## 2. Related Works

GFlowNets (Bengio et al., 2021) have emerged as a powerful framework for sampling diverse candidates proportional to a given reward, distinguishing them from traditional reinforcement learning (RL) approaches. They have been applied to molecular design (Jain et al., 2022), causal discovery (Deleu et al., 2022), and scheduling (Zhang et al., 2023). Unsupervised pretraining has been explored in RL to enhance sample efficiency (Liu & Abbeel, 2021; Kim et al., 2024) and in GFlowNets through outcome-conditioned training (Pan et al., 2023), which enables reward-free pretraining for adaptive downstream learning. Recent GFlowNet extensions incorporate goal-conditioning (Jain et al., 2022; Roy et al., 2023), allowing for controlled generative modeling, and multi-objective formulations (Jain et al., 2023), enabling trade-offs between competing molecular properties. These advances improve sample efficiency and targeted molecule generation.

RL-based molecular design has leveraged auto-regressive SMILES generation (Olivecrona et al., 2017; Popova et al., 2018) and graph-based approaches (You et al., 2018), while GFlowNets provide an alternative with improved diversity and exploration. For a comprehensive review, see (Bilodeau et al., 2022). A more detailed discussion of related works is provided in Appendix B.

## 3. Preliminaries

GFlowNets (Bengio et al., 2021) are generative models designed to sample structured objects from a state space $\mathcal{S}$ in proportion to a reward function $R(s_T)$ assigned to terminal states $s_T$. A model generates trajectories $\tau = (s_0, a_0, \ldots, s_T)$ by transitioning between states through actions, guided by a forward policy $P_F(s' \mid s)$. The key idea in GFlowNets is that they are trained to preserve a *flow* quantity, such that the total flow quantity entering a terminal state is equal to its unnormalized probability. This is achieved by aligning forward and backward policies $P_F$ and $P_B$, which can be understood as distributing flow forward and backwards through the state space. This induces a policy $P_F$ which at convergence yields $p_{\tau \sim P_F}(s_T) \propto R(s_T)$.

Existing extensions of GFlowNets that handle multiple, potentially conflicting objectives (Jain et al., 2023; Zhu et al.,

2024), along with the benefits of scalability to large action spaces and credit assignment makes them well-suited for molecular generation tasks; the primary focus of our work. We consider molecules as their topological graphs $G = (A, E, \mathcal{X})$ such that $\mathcal{X} \in \mathbb{R}^{n \times d}$ define d-dimensional atomic features for the $n$ nodes in $G$, $E \in \{0, 1\}^{b \times n \times n}$ is the edge-adjacency tensor for $b$ types of edges connecting $n$ nodes, and finally $A \in \{0, 1\}^{n \times n}$ is the adjacency matrix for $n$ nodes. A trajectory $\tau = (s_0, a_0), \ldots. (s_n, a_n)$ can be generated *de novo* from $s_0$ by sampling actions $a$ from a conditional policy $P_F(. \mid c; \theta)$, or by "deconstructing" a molecular graph $G = s_n$ via $P_B(. \mid c; \theta)$. States $s_i$ are partially constructed subgraphs, albeit we ensure that they remain valid molecular graphs throughout.

Our primary objective is to learn a model $\mathcal{G}_\theta = (P_F(; \theta), P_B(; \theta))$, such that the generated $G$ are chemically valid molecular graphs satisfying some molecular property conditional ranges $c$, and exhibiting sufficient diversity. We express this through a conditional reward $R(G|c)$ measuring how well $G$ satisfies $c$. We also want to be able to finetune these models to achieve drug discovery tasks with certain molecular property constraints, and establish the benefits of finetuning GFlowNets. Trajectory balance (TB; Malkin et al., 2022), being a standard GFlowNet objective, is used as the training method in this work. TB requires learning an additional parameter $\log Z_\theta$ which estimates the partition function. The loss is:

$$\mathcal{L}_{\text{TB}}(\tau) = \left( \log \left( \frac{Z_\theta \prod_{t=1}^{n} P_F(s_t | s_{t-1}; \theta)}{R(x) \prod_{t=1}^{n} P_B(s_{t-1} | s_t; \theta)} \right) \right)^2 \quad (1)$$

where trajectories $\tau = (s_0, s_1, \ldots, s_n)$ are such that $s_n = x$ is a fully constructed object.

## 4. Pretraining with Inexpensive Rewards

To construct molecular graphs via $P_F$, we design an action space with 5 action types: {addNode, addEdge, addNodeAttribute, addEdgeAttribute, stop}. The agent can add a node (heavy atom) to the graph, add an edge between two nodes, set a node's properties (e.g. its chirality), set a bond's properties (i.e. its bond order), or stop the trajectory. To deconstruct them via $P_B$, we use the deleterious actions: {deleteNode, deleteEdge, removeNodeAttribute, and removeEdgeAttribute}, which undo their constructive counterpart.

We use a graph neural network (Veličković et al., 2017), see §C.1, to parameterize the policies, using the GNN's invariances to produce per-node and per-edge logits. We are also careful to mask these logits such that the produced molecules always have valid valences (i.e. by design all states are convertible to RDKit molecules and SMILES). Since our method generates molecules atom by atom, we refer to the approach as **Atomic GFlowNet (A-GFN)**.

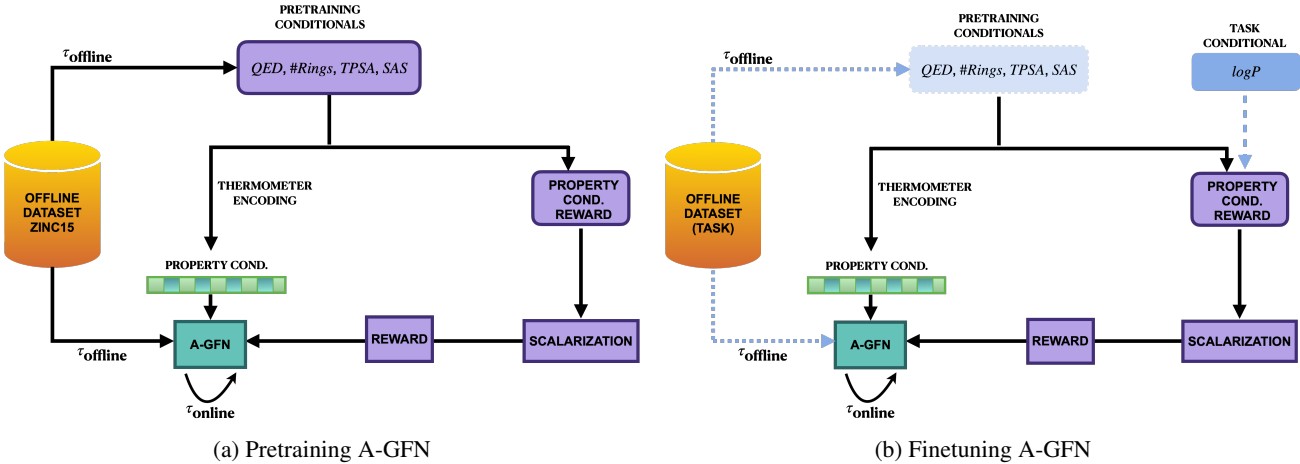

(a) Pretraining A-GFN          (b) Finetuning A-GFN

*Figure 1.* Illustration of the AGFN workflow, showing the overall process divided into two main parts: (a) Pretraining phase and (b) Finetuning phase.

## 4.1. Inexpensive Molecular Rewards

In the context of pre-training A-GFN models for molecular design, we utilize inexpensive molecular rewards—such as Topological Polar Surface Area (TPSA), Quantitative Estimate of Drug-likeness (QED), synthetic accessibility (SAS), and the number of five or six-membered rings in a molecule. These rewards are computationally cheap to evaluate and serve as proxies for more complex properties. We will later (§5) discuss finetuning models on more expensive and computationally intensive tasks, such as predicting binding affinity or toxicity (e.g., LD50), which are crucial for drug discovery but require significant computational resources or experimental data to assess accurately.

*Table 1.* Properties used as Pretraining conditionals in Figure 1 along with the probability distributions they are sampled from.

| Property | Distribution | |
|---|---|---|
| QED | $\mathcal{N}(X_{qed}, \sigma^2; 0.65, 0.80)$ | $U(0.65, 0.80)$ |
| #Rings | $\mathcal{N}(X_{\#rings}, \sigma^2; 1, 3)$ | $U(1, 3)$ |
| TPSA | $\mathcal{N}(X_{tpsa}, \sigma^2; 60, 100)$ | $U(60, 100)$ |
| SAS | $\mathcal{N}(X_{sas}, \sigma^2; 1, 3)$ | $U(1, 3)$ |

### 4.1.1. REWARD FUNCTION

To pre-train a goal-conditioned A-GFN for learning molecular properties $p \in P$, we define property-specific conditional ranges $c_p = (c_{low}, c_{high})$ which we want samples to belong to. We use the following goal-conditioned reward function for property $p$ and molecule $x$:

$$R_p(x|c_p) = \begin{cases} \frac{2-[d_p]^+}{2} \exp \frac{p_x - c_{low}}{\lambda} & \text{if } p_x < c_{low}, \\ \frac{2-[-d_p]^+}{2} \exp \frac{c_{high} - p_x}{\lambda} & \text{if } p_x > c_{high}, \\ \frac{2-[d_p]^+}{2} + \frac{d_p}{2} \frac{p_x - c_{low}}{c_{high} - c_{low}} & \text{otherwise.} \end{cases} \quad (2)$$

here, $\lambda$ controls the decay rate, $[x]^+ = \max\{0, x\}$, $c_{low}$ and $c_{high}$ are the lower and upper bounds for property $p$, and $d_p \in \mathbb{R}$ (typically $d_p \in \{-1, 0, 1\}$) represents the "preference direction" hyperparameter, indicating whether lower or higher property values within the range $c_p$ are preferred (for further details, see appendix D,E). Note that during training, we sample these ranges so that the model learns to be robust to inference-time queries (sec 4.1.2).

Drug discovery is an inherently multi-objective optimization (MOO) problem where the drug candidates are expected to simultaneously have several desired properties, such as a high drug-likeliness (QED), high SAS, and reasonably low TPSA, among other criteria. To implement this, we choose this standard multiplicative scalarization (Jain et al., 2023) of the multi-objective reward over the property set $P$:

$$R(x|c_{p_1}, ..., c_{p_{|P|}}) = \prod_{p \in P} R_p(x|c_p) \quad (3)$$

### 4.1.2. REWARD CONDITIONING A-GFN

To train our conditional model, we sample for every property $c_p = (c_{low}, c_{high})$ for each new training trajectory, using reasonable lower and upper bounds $c_{\min}$ and $c_{\max}$ of the molecular property ranges we care about, as well as extrema values $c_{\min}^*$, $c_{\max}^*$ of possible values we can imagine the property taking. This ensures that the model learns from a diversity of chemical regions relevant to drug discovery.

We use two types of training trajectories, online trajectories sampled from $P_F$, and offline trajectories whose terminal state is drawn from a dataset and then sampled backwards from $P_B$. For online trajectories, we draw $c_{low}, c_{high} \sim U(c_{\min}, c_{\max})$ uniformly (maintaining $c_{low} < c_{high}$), but with some probability $\epsilon$ instead draw from $U(c_{\min}^*, c_{\max}^*)$ to teach the model about unlikely but possible value ranges.

For offline trajectories ending in some state $x$, since we know each property $p_x$'s value, we center our sampling around them. We draw $c_{low}, c_{high} \sim \mathcal{T}(p_x, \sigma_p, c_{\min}, c_{\max})$ from $\mathcal{T}$ a truncated normal distribution whose $\sigma_p$ is a hyperparameter (again maintaining $c_{low} < c_{high}$). We also provide negative ($\approx 0$ reward) examples with some small probability $\epsilon$, for which with probability $1/2$ we set $c_{low} = c_{\min}$ and draw $c_{high} \sim U(c_{\min}, p_x)$, and otherwise set $c_{high} = c_{\max}$ and draw $c_{low} \sim U(p_x, c_{\max})$.

Finally, we encode $c_p$ via two thermometer encodings (Buckman et al., 2018) with bounds $c^*_{\min}, c^*_{\max}$.

### 4.2. Pretraining GFN with expert offline trajectories

Bengio et al. (2021) employ replay buffer-based off-policy training for GFlowNets to generate novel molecules. However, such online off-policy methods can suffer from high variance (Fedus et al., 2020; Vemgal et al., 2023) and a lock-in of suboptimal trajectories. This risk is high in sparse reward settings where the agent struggles to adequately explore rare, high-reward regions of the state space, resulting in slow convergence and suboptimal performance. To mitigate these challenges, we propose leveraging the vast amounts of readily available inexpensive and unlabelled molecular data to perform a hybrid online-offline off-policy pretraining of A-GFN. In this work we use the ZINC dataset (Sterling & Irwin, 2015) containing over 200M molecules from which we derive expert trajectories.

### 4.3. Balancing Exploration and Synthesis Feasibility

The primary benefit of molecular generative models with large explorative capacities such as A-GFN is their ability to navigate the almost arbitrarily large possibilities of chemical structures. On the other hand, ensuring that generated molecules lie close to the chemical space feasible for synthesis, particularly within the constraints of make-on-demand (MOD) libraries such as ZINC, is crucial for synthesis and *in vitro* validation in drug discovery. To balance these two seemingly conflicting goals, we introduce a regularization term in the pretraining objective. Specifically, we combine the exploration objective with a Maximum Likelihood Estimation (MLE) loss over the offline dataset, leading to the following regularized loss function:

$$\mathcal{L}(\tau) = \lambda_1 \mathcal{L}_{TB}(\tau) + \lambda_2 \mathcal{L}_{MLE}(\tau) \qquad (4)$$

$$where, \ \mathcal{L}_{MLE}(\tau) = -\prod_{t=1}^{n} \log P_F(s_t | s_{t-1}, \{c_p\}).$$

$\mathcal{L}_{TB}$ encourages exploration of the chemical space, while $\mathcal{L}_{MLE}$ encourages proximity to MOD libraries, and $\lambda_1, \lambda_2$ are hyperparameters controlling the trade-off between exploration and adherence to the MOD space (see Tab. 7).

With the online and offline trajectories described in §4.2

and molecular property rewards in §4.1, we optimize the A-GFN ($\mathcal{G}_\theta$) by minimizing Eq. (4) until convergence.

## 5. Finetuning

We investigate using pretrained A-GFN models ($\mathcal{G}_\theta$) and adapting them for downstream drug-discovery tasks, characterized by more challenging rewards. As usual, we initialize a new model with $\phi := \theta$.

We finetune $\mathcal{G}_\phi$ by using a new task-specific reward $R_{ext}$:

$$R_{ft}(x|c_{p_1}, ..., c_{p_{|P|}}) = R_{ext}(x) \prod_{p \in P} R_p(x) \qquad (5)$$

where $R_p(x)$ denotes the previous property-based rewards used during pretraining. Such a reward formulation ensures that the GFlowNet receives a high reward for generating molecules that simultaneously follow the desired molecular properties and are highly suitable for the downstream task. It should be noted that $Z_\theta$ in Eq. (1) is the normalization constant $\approx \sum_{x \in \mathcal{X}} R(x)$. While we do initialize our fine-tuned model with this value, we also recalibrate it to account for the updated reward $R_{ft}$, by adding some $\mathcal{N}(0, \sigma I)$ noise to $Z_\phi$ - a common strategy in finetuning (Yuan et al., 2023; Tong et al., 2022).

Finally, we finetune our models using the Relative Trajectory Balance objective (RTB; Venkatraman et al., 2024). RTB enables us to treat the density over objects $p_\theta(x|c)$ induced by $\mathcal{G}_\theta$'s $P_F$ as a *prior*, and to learn a posterior $p_\phi^{post}(x|c)$ such that $p_\phi^{post}(x|c) \propto p_\theta(x|c) R_{ft}(x|c)$. The RTB objective is:

$$\mathcal{L}_{RTB}(\tau; \phi) = \left( \log \left( \frac{Z_\phi \prod_{t=1}^{n} P_F(s_t|s_{t-1}; \phi)}{R(x) \prod_{t=1}^{n} P_F(s_t|s_{t-1}; \theta)} \right) \right)^2 \tag{6}$$

where $\phi$ is optimized and $\theta$ is fixed.

## 6. Experiments

We evaluate A-GFN's effectiveness during both pretraining and finetuning using comprehensive metrics such as novelty, diversity (Polykovskiy et al., 2020), uniqueness, validity, L1-distance, #modes, #scaffolds, #circles (Xie et al., 2021b), hit-ratio, and novel hit ratio (Guo & Schwaller, 2024b). Validity for A-GFNs is trivially 1 by construction and is not reported. **#Modes**: Measures the number of distinct, high-reward molecular clusters. A mode is defined as any molecule with reward $\geq 0.5$ that has Tanimoto similarity $<$ 0.5 to all previously counted modes.

**RW#C, RW#S, RWTD**: Reward-weighted circle count (RW#C), scaffold count (RW#S), and Tanimoto diversity (RWTD) capture structural diversity among high-reward molecules. Each is computed as the product of a diversity

metric and the top-K reward average: $RW - Metric = R_{topK} \times Metric$, typically with $K = 100$

**Normalized L1-dist**: This measures how far a generated molecule's properties deviate from their target ranges. For each property, the L1 distance is calculated by taking the absolute difference between the generated property value and the $10^{th}$ percentile value in desired range, and then normalized by the respective property range to ensure comparability across properties.

**Success Percent**: For a set of $N$ generated graphs $\{G\}_{i=1}^{N}$ and conditionals $C_{task}$ for given task, we define success percentage as

$$S_{\text{task}} = \frac{1}{N} \sum_{i=1}^{N} \left( \frac{1}{|\mathcal{C}_{\text{task}}|} \sum_{c \in \mathcal{C}_{\text{task}}} \mathbb{I}_c(G_i) \right) \times 100$$

where, $\mathbb{I}_c(G)$ counts the molecules within desired conditional ranges and defined as

$$\mathbb{I}_c(G) = \begin{cases} \mathbb{I}\left(|c(G) - c_{\text{low}}| \leq 0.1 \cdot |c_{\text{low}}|\right), & \text{if } d_p < 0 \\ \mathbb{I}\left(|c(G) - c_{\text{high}}| \leq 0.1 \cdot |c_{\text{high}}|\right), & \text{if } d_p > 0 \\ \mathbb{I}\left(c_{\text{low}} \leq c(G) \leq c_{\text{high}}\right), & \text{if } d_p = 0 \end{cases}$$

where, for any considered task, $c(G)$ is the the value of the conditional property $c$ for the molecular graph $G$, $c_{low}$ and $c_{high}$ are the lower and upper bounds of $c$, $d_p(c)$ indicates whether lower values ($< 0$), higher values ($> 0$), or values within a range ($= 0$) are preferred for that conditional.

**Hit Ratio**: Quantifies the percentage of molecules that meet the following criteria: docking score better than the median of known actives, QED > 0.5, and SA < 5 (Guo & Schwaller, 2024b).

**Novel Hit Ratio**: Restricts the hit ratio to molecules that are both unique and novel i.e., with maximum Tanimoto similarity < 0.4 to any training molecule (Lee et al., 2023b).

Full details of other standard metrics are provided in §E.

## 6.1. Tasks and Evaluation Criteria

We will first benchmark the pretraining phase of our proposed model, then benchmark its finetuning capabilities on well-understood exemplar tasks. Finally, we will evaluate our model on challenging practical tasks.
**Pretraining**: We compare the performance of A-GFN against fragment-based GFlowNets, conditioned on the same inexpensive molecular properties (sec 4.1). Our results show that A-GFN significantly outperforms fragment-GFN in exploring drug-like chemical space across multiple objectives.

**Finetuning**: We benchmark the pretrained A-GFN against A-GFN trained from scratch (task-trained A-GFN) on sev-

eral tasks from the TDC benchmark (Huang et al., 2022). To demonstrate the practical utility of A-GFN, we benchmark it against contemporary works in *de novo* binder generation, and lead optimization.

Following the setup in You et al. (2018), we focus on the following finetuning objectives:
*Property Optimization*: In this single objective problem, the goal is to generate molecules that maximize or minimize a specified physicochemical, structural, or binding property while ensuring diversity among the generated molecules. In this unconstrained optimization setting, we compare A-GFN finetuned with TB and RTB to the task-trained A-GFN
*Property Targeting*: In this multi-objective problem, we generate molecules that satisfy property constraints different from those used during pretraining, while ensuring structural distinction from the pretraining set.
*Property Constrained Optimization*: This problem requires generating molecules that simultaneously meet property optimization and property targeting criteria, i.e. molecules must lie within the specified property ranges while also maximizing (or minimizing) the targeted property.

**Binding Tasks**: To demonstrate the practical applicability of A-GFN, we benchmark it against state-of-art models for *de novo* molecular generation, and perform lead optimization. In *de novo* generation, we evaluate A-GFN on its ability to generate high-affinity molecules for specified protein targets. For *lead optimization*, we assess its capability to refine initial molecular scaffolds and improve their binding affinity, while adhering to synthetic accessibility and other physicochemical constraints.

## 6.2. Baselines

We evaluate A-GFN against a diverse set of molecular generative state-of-the-art baselines. To be consistent with the atomic resolution of A-GFN, we include MolDQN (Zhou et al., 2019) and GraphMCTS (Jensen, 2019) (both of which construct molecular graphs atom-by-atom), and REINVENT 4 (Loeffler et al., 2024), a widely adopted baseline in the pharmaceutical industry known for its multi-objective molecular optimization capabilities and seamless integration with in-silico screening tools (which uses a character-level RNN for SMILES, i.e. also atom-level). To assess the benefits of A-GFN over fragment-based approaches, we use an unconditional fragment-based GFlowNet baseline, trained from scratch on the optimization task. For *de novo* molecular generation, we compare against the baselines considered in Guo & Schwaller (2024b), which leverage diverse generative paradigms to model chemical space.

## 6.3. Pretraining

To create a robust initialization for downstream molecular generation tasks, we pretrain A-GFN using a hybrid online-

offline off-policy learning strategy, where each minibatch consists of half offline trajectories sampled from an expert dataset and half on-policy trajectories. The reward is conditioned on molecular properties—TPSA, QED, SAS, and ring count—constrained within predefined target ranges. To benchmark fragment-based GFlowNets, we adapt the purely online training paradigm of (Malkin et al., 2022). Both methods are conditioned on the same properties and ranges.

Empirical results indicate that A-GFN learns a broad distribution over valid and synthetically accessible molecules while significantly improving scaffold diversity, covering nearly twice as many distinct scaffolds as the fragment-based GFlowNet (Tab. 2). Furthermore, molecules sampled from the pretrained A-GFN are chemically valid, novel, and adhere to specified property constraints without directly replicating structures from ZINC250K (Fig. 3). Complete details on molecular constraints, training dynamics, and benchmarking are provided in Appendix C.

*Table 2.* Comparison of pretraining A-GFN and fragment-based GFlowNet on the same molecular property conditionals.

| Method | #Modes | #scaffold (n=200) | Div | Uniq | Nov | Valid |
|--------|--------|-------------------|-----|------|-----|-------|
| Frag. GFN | 0 | 113 | 0.002 | 0.001 | 0.97 | 0.99 |
| A-GFN | **1252** | **196** | **0.88** | **0.96** | **1.0** | 1.0 |

### 6.4. Finetuning

We investigate a number of tasks: property optimization, property targeting, and property constrained optimization. Each task corresponds to a set of conditionals and a reward function, detailed in subsequent sections. For tasks where some small labeled dataset is accessible, we investigate the benefits of finetuning A-GFNs off-policy with that data. Following You et al. (2018), we consider two structural tasks, molecular weight (M.W.), and logP (partition coefficient for drugs' water-to-octanol concentration; Sangster, 1997). We also consider some standard drug-discovery tasks where rewards are based on empirical models such as the ADME-T (absorption, distribution, metabolism, excretion, and toxicity) tasks of TDC (Huang et al., 2021). Using these, we aim to show that finetuning a pretrained A-GFN achieves the task's objectives more quickly than training a new A-GFN from scratch (Task Trained A-GFN). Unless stated otherwise, all models are finetuned for 24 hours using the RTB training objective on a single V100 GPU. Results are based on 1000 samples for all tasks except binding, where 3000 samples are used for consistency with prior work (Guo & Schwaller, 2024b).

### 6.4.1. PROPERTY OPTIMIZATION

In this task, the goal is to generate novel molecules that minimize (or maximize) specific physicochemical properties. While previous works have commonly benchmarked their models on QED (Jeon & Kim, 2020; Zhou et al., 2019; You et al., 2018), we exclude QED from our evaluation, as our pretrained A-GFN is already optimized for it. Instead, in 3 independent tasks, we focus on molecular weight (M.W.), logP, and toxicity respectively. We will consider harder tasks and additional baselines in later sections.

As expected, for the simple tasks of M.W. and logP optimization, the finetuned A-GFN has a higher success rate than the task-trained A-GFN, as indicated by higher top-k rewards and an increased number of distinct modes. However, for more complex tasks such as LD50 prediction, both finetuning and task training struggle (Table 3), suggesting that a more sophisticated reward scheme beyond direct task rewards is necessary.

Interestingly, in these relatively simple single-objective optimization tasks, finetuning A-GFN using the TB objective outperforms using the RTB objective (Table 10). This can be attributed to the inherent constraints imposed by the RTB objective, which anchors the policy to the pretrained prior designed for drug-likeness. While beneficial for maintaining chemically valid and synthesizable molecules, this constraint limits exploration of non-drug-like solutions that might satisfy the optimization objective more easily.

*Table 3.* Comparison of finetuned A-GFN with A-GFN trained from scratch for logP, molecular weight, and toxicity optimization tasks. Detailed comparaision in table 10

| Task | Method | #Modes↑ | Div↑ | L1-Dist↓ | Nov↑ | Rew (top-100) |
|------|--------|---------|------|----------|------|---------------|
| MW | A-GFN Task trained | 0 | 0.12 | 4.56 | 1.0 | $< 10^{-5}$ |
| | **A-GFN Finetuned** | **992** | **0.91** | **0.08** | **1.0** | **0.98** |
| logP | A-GFN Task trained | 10 | 0.16 | 2.65 | 1.0 | $< 10^{-5}$ |
| | **A-GFN Finetuned** | **994** | **0.84** | **0.06** | **1.0** | **0.98** |
| Toxicity | A-GFN Task trained | 0 | 0.02 | 1.56 | 1.0 | $< 10^{-5}$ |
| | **A-GFN Finetuned** | **0** | **0.16** | **0.75** | **1.0** | **0.18** |

### 6.4.2. PROPERTY TARGETING

Here, we the goal is to generate molecules that satisfy an altered range for a specific molecular property constraint used during the pretraining phase. This demonstrates the model's capacity for finetuning while adapting to modified property ranges. We call this adaptation Dynamic Range Adjustment (DRA). Compared to task-training, finetuning an A-GFN leads to significantly faster convergence, even when property ranges are altered. We report results on modifying the TPSA property while keeping the constraints for other molecular properties as in pretraining. To assess generalization to new property constraints, we shift the TPSA range from [60, 100] (pretraining) to [40, 60] and [100, 120]. Finetuned A-GFN consistently surpasses training from scratch in both convergence speed and performance, excelling in high-reward mode discovery (#modes), diversity, and success rate, with notable gains under the RTB objective (Tab. 4, 11).

The disparity in performance between the TB and RTB objectives becomes increasingly pronounced in this MOO problem (Tab. 11). Finetuning without the RTB objective exhibits catastrophic forgetting, wherein the model loses the ability to generate chemically valid, drug-like molecules learned during pretraining. Although in some cases, the model eventually relearns these skills, that process is inefficient and slow. In other words, while extended training improves the TB objective by progressively aligning the learned policy with the reward signal, overtraining the RTB objective degrades its performance.

In the RTB formulation, the updated posterior distribution is proportional to the product of the prior and the reward function, i.e., $p_\theta(x) \propto p_0(x)R(x)$, where $p_0(x)$ is the pretrained prior distribution and $R(x)$ is the reward function. As training progresses, this posterior collapses onto high-reward regions while maintaining proximity to the prior, thereby constraining the effective solution space. Consequently, longer training under the RTB objective produces high-reward molecules but at the cost of reduced diversity and uniqueness in the generated samples (Fig. 10). Thus, a trade-off between diversity and optimization efficiency must be carefully balanced as a function of training duration when using the RTB objective.

The ability of the finetuned A-GFN to adapt to modified property constraints highlights the robustness of the pretrained $\mathcal{G}_\theta$ model, which has not merely memorized specific property ranges but has learned about the state space's structure. This adaptability is crucial in real-world drug discovery applications, where molecular property requirements often shift during the drug discovery process.

| DRA | Method | #Modes | HV↑ | RWTD↑ | Success % | Nov↑ | Rew↑ (top-100) |
|---|---|---|---|---|---|---|---|
| High TPSA | A-GFN Task Trained | 0 | 0.16 | 0.07 | 0 | 0.24 | 0.15 |
| | **A-GFN FT w/ RTB** | 147 | 1.0 | 0.68 | 94.5 | 0.65 | 1.0 |
| Low TPSA | A-GFN Task Trained | 0 | 0.03 | 0.03 | 0 | 0.02 | 0.03 |
| | **A-GFN FT w/ RTB** | 79 | 1.0 | 0.56 | 94.6 | 0.80 | 1.0 |

*Table 4.* Comparing the effectiveness of finetuned A-GFN over A-GFN trained from scratch for TPSA adjusted property targeting objective. Detailed comparision in table 11

### 6.4.3. PROPERTY CONSTRAINED OPTIMIZATION

For this comprehensive benchmarking task, the aim is to generate molecules that optimize specific target properties—such as M.W., logP, or toxicity, *while* preserving essential drug-like characteristics learned during pretraining. We evaluate two key finetuning paradigms: conditionals-preserved finetuning and Dynamic Range Adjustment (DRA), demonstrating that finetuning with RTB consistently improves performance across all ADME-T

| | Method | Chem. Space Coverage | Diversity | MOO metrics | | | | | | Reward |
|---|---|---|---|---|---|---|---|---|---|---|
| | | | | | | L1-dist ↓ | | | | |
| | | #modes | RWTD | HV | TPSA | #Rings | SAS | QED | Task | (Top-100) |
| Preserved | MolDQN | 0 | 0.255 | 0.582 | 0 | 0.353 | 0.14 | 0.096 | 0.175 | 0.337 |
| | Reinvent | 139 | 0.545 | 0.835 | 0 | 0 | 0 | 0.233 | 0.106 | 0.759 |
| | Frag GFlowNet | 51 | 0.461 | 0.820 | 0 | 0.13 | 0.056 | 0.204 | 0.135 | 0.536 |
| | Graph MCTS | 3 | 0.347 | 0.713 | 0 | 0.524 | 0.055 | 0.17 | 0.233 | 0.404 |
| | AGFN Tasktrained | 0 | 0.000 | 0.0001 | 0.065 | 0.956 | 2.862 | 1.000 | 0.446 | $< 10^{-5}$ |
| | AGFN Finetuned | 1 | 0.040 | 0.677 | 0.086 | 0.827 | 2.047 | 3.622 | 0.311 | 0.0441 |
| | AGFN FT w/ data | 419 | 0.735 | 0.991 | 0.031 | 0.417 | 0.172 | 0.266 | 0.187 | 0.8547 |
| | AGFN FT w/ RTB | 708 | **0.793** | **0.996** | 0.011 | 0.069 | 0.029 | 0.174 | 0.126 | **0.9804** |
| | AGFN FT w/ RTB + data | 351 | 0.716 | 0.983 | 0.026 | 0.364 | 0.191 | 0.266 | 0.259 | 0.8333 |
| Low TPSA | MolDQN | 0 | 0.189 | 0.517 | 0 | 0.514 | 0.292 | 0.106 | 0.301 | 0.244 |
| | Reinvent | 162 | 0.578 | 0.889 | 0 | 0.032 | 0 | 0.237 | 0.141 | 0.776 |
| | Frag GFlowNet | 106 | 0.524 | 0.88 | 0 | 0.093 | 0.005 | 0.168 | 0.169 | 0.622 |
| | Graph MCTS | 13 | 0.392 | 0.79 | 0 | 0.493 | 0.036 | 0.16 | 0.234 | 0.456 |
| | AGFN Tasktrained | 0 | 0.000 | 0.0001 | 0.351 | 1.235 | 0.321 | 0.654 | 1.235 | 0.000 |
| | AGFN Finetuned | 0 | 0.027 | 0.534 | 0.181 | 0.961 | 1.996 | 2.918 | 0.304 | 0.029 |
| | AGFN FT w/ data | 317 | 0.684 | 0.995 | 0.080 | 0.432 | 0.182 | 0.278 | 0.250 | 0.796 |
| | AGFN FT w/ RTB | 479 | **0.753** | **0.997** | 0.079 | 0.154 | 0.039 | 0.182 | 0.215 | **0.939** |
| | AGFN FT w/ RTB + data | 112 | 0.556 | 0.986 | 0.145 | 0.331 | 0.288 | 0.358 | 0.485 | 0.645 |
| High TPSA | MolDQN | 0 | 0.012 | 0.116 | 0 | 0.901 | 1.098 | 2.114 | 0.252 | 0.013 |
| | Reinvent | 102 | 0.463 | 0.748 | 0 | 0.007 | 0 | 0.131 | 0.091 | 0.674 |
| | Frag GFlowNet | 0 | 0.237 | 0.652 | 0 | 0.22 | 0.235 | 0.46 | 0.206 | 0.273 |
| | Graph MCTS | 0 | 0.165 | 0.550 | 0 | 0.554 | 0.286 | 1.011 | 0.176 | 0.189 |
| | AGFN Tasktrained | 0 | 0.000 | 0.002 | 1.360 | 1.654 | 0.032 | 0.900 | 1.220 | $< 10^{-5}$ |
| | AGFN Finetuned | 0 | 0.031 | 0.454 | 0.288 | 0.911 | 2.122 | 3.941 | 0.275 | 3.41E-02 |
| | AGFN FT w/ data | 115 | 0.564 | **0.998** | 0.787 | 0.419 | 0.190 | 0.279 | 0.173 | 0.654 |
| | AGFN FT w/ RTB | 490 | **0.741** | 0.994 | 0.137 | 0.206 | 0.031 | 0.389 | 0.153 | **0.936** |
| | AGFN FT w/ RTB + data | 260 | 0.667 | 0.994 | 0.468 | 0.355 | 0.173 | 0.316 | 0.202 | 0.776 |

*Table 5.* Comparing the effectiveness of finetuned A-GFN over A-GFN trained from scratch and other baselines for property-constrained optimization objective for a representative logP task. For a more extensive comparision, see §F.4

tasks from TDC (Cell Effective Permeability, Lipophilicity, Solubility, Plasma Protein Binding Rate, Hepatocyte Clearance, Microsomal Clearance and Acute Toxicity) as shown in Tables 12 through 20.

In the conditionals-preserved finetuning setup, we retain the same conditional property ranges ($c_p$) used during pretraining, setting the task's $d_p$ in accordance to table 9, which directs the model to minimize the target property. The desired ranges for these task properties are set between the $25^{th}$ percentile of the ZINC dataset and a predetermined maximum threshold (see Tab. 8 and 9 for specifics).

The finetuned A-GFN significantly outperforms the A-GFN trained from scratch, achieving superior results within the same computational budget. In the dynamic range adjustment scenario, we modify one of the pretraining conditionals (TPSA) by shifting its range from $60 \le TPSA \le 100$ to both lower ($40 \le TPSA \le 60$) and higher ($100 \le TPSA \le 120$) ranges. In this case, the finetuned A-GFN again surpasses its non-pretrained counterpart, demonstrating faster convergence and higher success rates (Tab.5). We also explore the hybrid online-offline finetuning approach, with and without the RTB objective, where A-GFN lever-

ages offline task-specific data, with the desired $c_p$ similar to pretraining. In some tasks, hybrid finetuning with RTB outperforms other approaches for the metrics shown in the tables referenced above. However, this performance gain is accompanied by an inductive bias toward the data distribution of the offline fine-tuning dataset, particularly when its support is limited. Even though these are vast chemical spaces to be explored, evident through the number of modes metric, we also observe that a finetuned A-GFN with RTB significantly outperforms all baselines, by consistently achieving greater chemical diversity (RWTD metric).

For example, Table 13 shows results from a challenging molecular generation task where the goal is to generate molecules with logP≈1.5 while maintaining the conditional properties outlined in Tab. 9. The inherent difficulty of this task is underscored by the fact that only 0.002% of molecules in the ZINC250K dataset meet these stringent criteria. Despite the challenge, A-GFN finetuned with the RTB objective achieves better metrics than other baselines. This underscores the potential of hybrid online-offline finetuning to unlock otherwise inaccessible regions of the chemical landscape, offering a promising strategy for tackling difficult-to-sample molecular spaces.

Results from the experiments in the DRA paradigm, where we modify one of the conditional properties like TPSA, reinforce the fact that A-GFN with RTB enables faster adaptability when transitioning to new property distributions. All our results establish RTB as an effective strategy for constrained molecular property optimization, consistently reaching greater diversity, and improved adaptability when compared to other previous state-of-the-art baselines such as MolDQN, REINVENT, and Graph MCTS.

# 7. Case Study

In this section, we explore the generation of drug-like molecules by finetuning the pretrained A-GFN prior under a property constrained optimization setting for optimizing docking scores–an established *in silico* drug discovery proxy for protein-ligand binding. We use 5 targets widely adopted in recent studies, namely *fa7, barf, 5h1b, parp1,* and *jak2* (Guo & Schwaller, 2024b). Generating molecules suitable for docking is a challenging task, as valid 3D conformations must be realizable for accurate scoring. We find that models trained from scratch struggle to generate molecules with viable conformers that can be successfully docked. As such, we focus exclusively on finetuning the pretrained A-GFN prior, rather than training a model from scratch for each target.

## 7.1. *De novo* Generation

Following prior works (Lee et al., 2023b;a), we generate molecules for five protein targets, optimizing docking score (DS), QED, and SAS. Similar to Saturn (Guo & Schwaller, 2024b), benchmarking is conducted with QuickVina2 (Alhossary et al., 2015), and we report hit ratio and novel hit ratio (§E). Finetuning employs DRA-based property-constrained optimization with modified constraints on #rings and $d_p = 1$.

We evaluate two A-GFN priors: (i) trained solely on offline data (Enamine 6.75M, 18 days, 4 A100-40GB GPUs) and (ii) a hybrid online-offline prior (ZINC250K, 12 days, same hardware). With both priors, a high inverse-temperature finetuned A-GFN ($\beta \gg 1$) matches or surpasses Saturn in hit ratio (Tab. 21). However, under a strict novelty constraint (Tanimoto similarity $\leq 0.4$), the offline-only prior underperforms in novel hit ratio. This arises because RTB finetuning keeps the policy anchored to its prior, biasing generation toward known scaffolds.

For applications prioritizing structural novelty, finetuning the hybrid prior significantly improves exploration while maintaining competitive hit rates, outperforming all baselines (Tab. 6, 22). This highlights the benefit of integrating offline expertise with online exploration to enhance molecular diversity while preserving biological relevance. Full experimental details are provided in Appendix F.5.

*Table 6.* Percentage improvement by A-GFN finetuning of the hybrid prior over the best-performing method reported in (Guo & Schwaller, 2024b). Detailed benchmarking results are provided in Tables 21 and 22.

| Target | Percent Improvement | |
|---|---|---|
| | Hit Ratio | Novel Hit Ratio |
| parp1 | 31.72 | 37.66 |
| fa7 | 148.35 | 34.73 |
| 5ht1b | 43.59 | 35.86 |
| braf | 63.00 | 188.29 |
| jak2 | 60.98 | 54.29 |

## 7.2. Lead Optimization

In lead optimization, we modify a well-performing compound to enhance its properties. Here we consider improvement of docking score (DS) as an example. This score, while a common metric, is biased toward compounds with high molecular weight and lipophilicity (Carta et al., 2007; García-Ortegón et al., 2022), which often results in large, hydrophobic molecules with poor ADME-T properties and off-target effects. To address this, we optimize docking scores alongside ligand efficiency (Hopkins et al., 2014), defined as $-DS/\#HeavyAtoms$, while ensuring structural similarity to the lead compounds to preserve experimentally validated properties.

We use the Bemis-Murcko scaffold of the lead ($\mathcal{B}$) as the seed during finetuning ($s_0 = \mathcal{B}$) with RTB. We do not

change the property ranges. Through atom-wise decoration of $\mathcal{B}$ we improve the docking score and ligand efficiency for each of the 5 targets considered in Sec. 7.1. The best-docked compound from Zinc250K for each target is selected as the lead. Top-2 optimized results per target are shown in Fig. 11.

## 8. Conclusion

In this work, we introduced Atomic GFlowNets (A-GFN), an application of the GFlowNet framework that leverages atoms as fundamental building blocks to explore molecular space. By shifting the action space from predefined molecular fragments to individual atoms, A-GFN is able to explore a much larger chemical space than its fragment-based counterpart, enabling the discovery of more diverse and pharmacologically relevant molecules. To address the challenges posed by the vastness of this atomic action space, we propose a pretraining strategy using datasets of drug-like molecules. This off-policy pretraining approach conditions A-GFN on informative but computationally inexpensive molecular properties such as drug-likeness, topological polar surface area, and synthetic accessibility, allowing it to effectively explore regions of chemical space that are more likely to yield viable drug candidates.

In comparison with models trained from scratch, we show that models pretrained with both expert trajectories and self-generated online samples lead to improved diversity and novelty, as well as a lower amortized computational cost.

However, akin to challenges in finetuning RL models (Wolczyk et al., 2024), A-GFN is prone to catastrophic forgetting of its pretraining-phase knowledge when extensively finetuned. This highlights the utility of methods like relative trajectory balance (Venkatraman et al., 2024) that regularize the finetuning process, ensuring that the policy stays anchored to a priort while still adapting to task-specific goals.

While pretraining has been the hallmark of other generative frameworks (Brown, 2020), we believe that this work is the first to demonstrate a large scale, useful application of pretraining within the GFlowNet framework. Some obvious next steps thus include a deeper analysis of the scaling behaviors of this pretraining phase (Kaplan et al., 2020), in terms of amounts of *expert* data (of which there are potentially vast quantities in chemistry), of the size of models, and of the overall length of the pretraining phase. Indeed, thanks to its online/on-policy pretraining component, the model has, in principle, access to its *entire* state space. This means such models could in principle learn about the (simple) properties of all accessible compounds, given enough time, and become powerful priors to draw upon.

## Impact Statement

This work introduces A-GFNs, a novel framework for atom-level *de novo* molecular generation using GFlowNets. From a broader perspective, A-GFNs contribute to the development of scalable and flexible molecular design tools, potentially expediting early-stage drug discovery and lead optimization. The ability to generate diverse and pharmacologically relevant molecules can aid in identifying candidate compounds for underexplored or neglected diseases. However, any advances in molecular generation technology, including A-GFNs, must be coupled with careful ethical oversight to ensure responsible use in pharmaceutical research.

## Acknowledgments

This research was enabled by funds provided by a Canadian Institutes of Health Research (CIHR) doctoral award (FRN: FBD-187593) to MP. MP would like to further acknowledge numerous helpful discussions with Dr. Hong Wang. GS would like to acknowledge the support from the Canada CIFAR AI Chair Program and from the Canada Excellence Research Chairs (CERC) Program. AC would like to acknowledge Natural Sciences and Engineering Research Council of Canada (NSERC) Discovery Grant (RGPIN-2024-04153). We also thank the Mila cluster, Recursion Pharma compute cluster, and Compute Canada for providing computational resources.

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

# Appendix

## A. Markov Decision Process

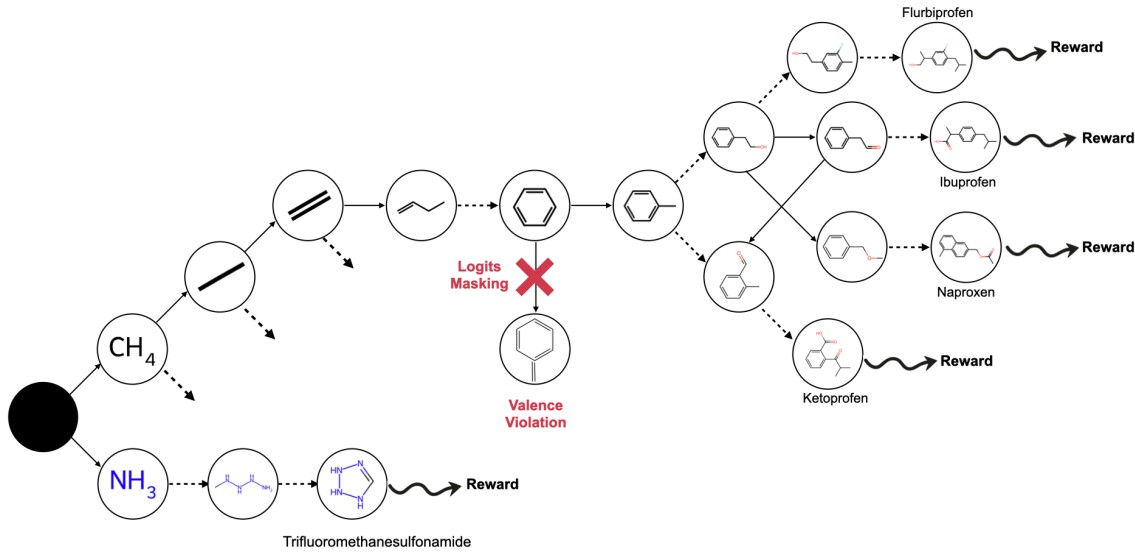

*Figure 2.* MDP governing the construction of molecules with A-GFN.

## B. Related Works

### B.1. GFlowNet

GFlowNets were introduced by Bengio et al. (2021) as a framework for training energy-based generative models that learn to sample diverse candidates in proportion to a given reward function. Unlike traditional Reinforcement Learning (RL) methods, which focus on maximizing rewards through a sequence of actions, GFlowNets aim to generate samples with probabilities proportional to their associated rewards. This distinction allows GFlowNets to explore a broader solution space, facilitating the discovery of novel, high-quality, and diverse objects across various domains. Recent works have been conducted along a multitude of directions, some focusing on theoretical aspects, such as connections to variational methods (Zhang et al., 2022; Malkin et al., 2023), and some focused on improved training methods for better credit assignment and sample efficiency (Malkin et al., 2022; Bengio et al., 2023). Due to their flexibility, GFlowNets have also been successfully applied to different settings such as biological sequences (Jain et al., 2022), causal discovery (Deleu et al., 2022; Atanackovic et al., 2023), discrete latent variable modeling (Hu et al., 2023), and computational graph scheduling (Zhang et al., 2023).

### B.2. Unsupervised pretraining in RL and GFlowNets

Pretraining models in machine learning has virtually become the default way to obtain powerful models (Brown, 2020; Kalapos & Gyires-Tóth, 2024). Specifically, unsupervised pretraining in reinforcement learning (RL) has emerged as a promising strategy to enhance data efficiency and improve agent performance across various tasks. Recent works have explored different methodologies to leverage unsupervised interactions with the environment before finetuning on specific objectives. For instance, Liu & Abbeel (2021) introduced Active Pre-Training, a reward-free pre-training method that maximizes particle-based entropy in a contrastive representation space, achieving human-level performance on several Atari games and significantly enhancing data efficiency in the DMControl suite. Similarly, Mutti et al. (2021) addressed unsupervised RL in multiple environments, proposing a framework that allows for pre-training across diverse scenarios to improve the agent's adaptability. Additionally, the Unsupervised-to-Online RL framework (Kim et al., 2024) was developed, which replaces domain-specific offline RL with unsupervised pre-training, demonstrating that a single pre-trained model

can be effectively reused for multiple downstream tasks, often outperforming traditional methods. Recent advancements in GFlowNets have also introduced unsupervised pre-training strategies, such as the outcome-conditioned GFlowNet (Pan et al., 2023), which enables reward-free pre-training by framing the task as a self-supervised problem. This approach allows GFlowNets to learn to explore the candidate space and adapt efficiently to downstream tasks, showcasing the potential of unsupervised pre-training in enhancing the performance of generative models in molecular design.

### B.3. Goal-Conditioned and Multi-Objective Gflownets

GFlowNets have been extended to goal-conditioned and multi-objective frameworks, enhancing their applicability in complex generative tasks. Goal-conditioned GFlowNets enable the generation of diverse outputs tailored to specific objectives, improving sample efficiency and generalization across different goals, as demonstrated by Jain et al. (2022) and further refined through methods like Retrospective Backward Synthesis (He et al., 2024) to address sparse reward challenges. Roy et al. (2023), impose hard constraints through focus regions as a goal-design strategy, making it comparable to goal-conditioned reinforcement learning (Schaul et al., 2015). Additionally, the development of multi-objective GFlowNets (Jain et al., 2023) allows for simultaneous optimization of multiple criteria, providing practitioners with greater control over the generative process and the ability to explore the Pareto fronts of trade-offs between competing objectives.

### B.4. Molecule Generation

To contextualize our approach, we provide a brief review of prior research using atom-based vocabularies in molecular generative modeling. Reinforcement learning (RL) studies optimal decision-making methodologies to maximize cumulative rewards. Given the shared notion of object-constructing Markov decision policies between GFlowNets and RL, we focus on the latter literature for molecule generation. For a comprehensive review of methods in molecule generation, we point the readers to the work of Bilodeau et al. (2022). RL is frequently employed in de novo design tasks due to its capability to explore chemical spaces beyond the compounds present in existing datasets. Moreover, RL allows for targeted molecule generation for constrained property optimization. Early works focused on auto-regressive generation of SMILES within an RL-loop for molecular property optimization (Olivecrona et al., 2017; Popova et al., 2018; Goel et al., 2021). MolDQN (Zhou et al., 2019) employs deep Q-Networks and multi-objective molecular properties for scalarization of rewards to generate 100% valid molecules without pretraining on a dataset. You et al. (2018) and Atance et al. (2022) employed graph neural networks to generate molecular graphs. These are trained on offline molecular datasets as well as through policy-gradient updates to ensure that the generated molecules adhere to specified property profiles.

## C. Pretraining Setup

For our pretraining, we utilize ZINC250K, a curated subset of the ZINC database, which consists of 250,000 commercially accessible drug-like compounds drawn from over 37 billion molecules available in ZINC (Tingle et al., 2023).

The primary goal during pretraining is to optimize the A-GFN for generic yet critical drug-like properties: TPSA, QED, SAS, and the number of rings. The desired ranges for these properties are enumerated in Tab.1. These properties are chosen based on their established relevance in guiding molecular design towards compounds with desirable pharmacokinetic and pharmacodynamic profiles, following the framework of (Wellnitz et al., 2024). By optimizing for these properties, we aim to equip A-GFN with the capability to generate molecules that strike a balance between drug-likeness and structural diversity.

We restrict atom types to a core set commonly found in drug-like molecules: $C, S, P, N, O, F, Cl, Br, I$, and implicit hydrogen ($H$). This ensures that the generated molecules are synthetically relevant and pharmacologically plausible, avoiding rare or exotic atom types that are less likely to lead to viable drug candidates.

For benchmarking fragment-based GFlowNets, we adapt the purely online training setup proposed in Jain et al. (2023), conditioning it on the same molecular properties as A-GFN. To align with our A-GFN setup, we generate a fragment vocabulary from the BRICS decomposition of ZINC250K, selecting the 73 (following (Bengio et al., 2021)) most common fragments. This equips the fragment-based GFlowNet with a diverse and representative set of building blocks for molecular generation.

Through pretraining, we observe that A-GFN effectively adapts to the specified property ranges while maintaining high molecular diversity and uniqueness. All molecules generated by A-GFN are valid, adhering to chemical rules, and novel, as they do not replicate any molecules in the ZINC250K dataset (Fig.3).

For the same set of property conditionals, A-GFN demonstrates superior chemical scaffold exploration compared to the fragment-based GFlowNet, covering nearly twice as many distinct scaffolds (Tab.2). While the fragment-based method has a higher success rate and better control over specific molecular properties, A-GFN excels in uniqueness and novelty, making it a more powerful tool for exploring uncharted chemical space in drug discovery. This highlights the model's capacity to explore novel regions of chemical space while adhering to fundamental molecular design principles.

Pretraining was done on 4 Nvidia A100-40G GPUs for 12 days.

### C.1. Architecture

We use a graph neural network that is mainly a graph attention network (Veličković et al., 2017), using specifically the variant proposed by Shi et al. (2020b). To help the model with counting-like behavior, we also include convolutional layers with additive aggregation, following Li et al. (2020), whose output is concatenated with the input of each graph attention layer.

*Table 7.* Pretraining and finetuning setups and corresponding hyperparameters

| Hyperparameter | Pretraining Values | FT($\pm$ RTB) Values | FT ($\pm$ RTB) w/data Values |
|---|---|---|---|
| max_num_iter | 500,000 | | |
| bootstrap_own_reward | FALSE | | |
| random_seed | 1,428,570 | | |
| beta | 96 | 64 | |
| OOB_percent | 0.1 | | |
| zinc_rad_scale ($\lambda$) | 1 | | |
| gfn_batch_shuffle | FALSE | | |
| reward_aggregation | mul | | |
| sampling_batch_size | 2048 | 1024 | |
| training_batch_size | 64 | | |
| learning_rate | 1.00e-04 | | |
| online_offline_mix_ratio | 0.5 | | |
| num_workers | 8 | 2 | |
| gfn_loss_coeff ($\lambda_1$) | 0.04 | - | 0.04 |
| MLE_coeff ($\lambda_2$) | 20 | - | 20 |
| num_emb | 128 | | |
| num_layers | 8 | | |
| num_mlp_layers | 4 | | |
| num_heads | 2 | | |
| i2h_width | 1 | | |
| illegal_action_logreward | -512 | | |
| reward_loss_multiplier | 1 | | |
| weight_decay | 1.00e-08 | | |
| num_data_loader_workers | 8 | | |
| momentum | 0.9 | | |
| adam_eps | 1.00e-08 | | |
| lr_decay | 20,000 | | |
| Z_lr_decay | 20,000 | | |
| clip_grad_type | norm | | |
| clip_grad_param | 10 | | |
| random_action_prob | 0.001 | | |
| random_stop_prob | 0.001 | | |
| num_back_steps_max | 25 | | |
| max_traj_len | 40 | | |
| max_nodes | 45 | | |
| max_edges | 50 | | |
| tb_p_b_is_parameterized | TRUE | | |
| num_thermometer_dim | 16 | | |
| sample_temp | 1 | | |
| checkpoint_every | 1,000 | 500 | |
| Z_learning_rate | 1e-3 | | |

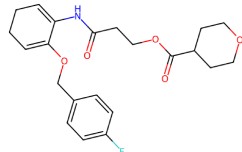

QED: 0.67, TPSA: 73.9,
Rings: 3, SAS: 2.79

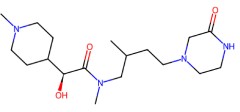

QED: 0.65, TPSA: 76.1,
Rings: 2, SAS: 3.65

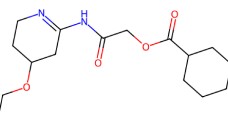

QED: 0.79, TPSA: 77.0,
Rings: 2, SAS: 3.13

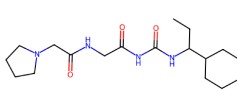

QED: 0.65, TPSA: 90.5,
Rings: 2, SAS: 2.76

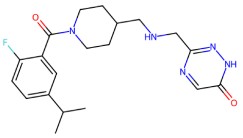

QED: 0.65, TPSA: 84.2,
Rings: 1, SAS: 1.96

QED: 0.54, TPSA: 70.7,
Rings: 2, SAS: 3.20

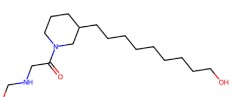

QED: 0.38, TPSA: 72.8,
Rings: 1, SAS: 2.86

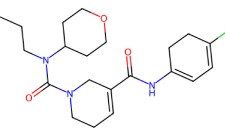

QED: 0.76, TPSA: 61.9,
Rings: 3, SAS: 3.40

QED: 0.79, TPSA: 91.0,
Rings: 3, SAS: 2.67

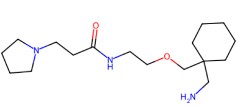

QED: 0.63, TPSA: 67.6,
Rings: 2, SAS: 2.45

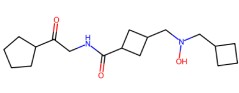

QED: 0.67, TPSA: 69.6,
Rings: 3, SAS: 2.87

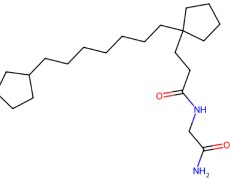

QED: 0.46, TPSA: 72.2,
Rings: 2, SAS: 2.54

QED: 0.70, TPSA: 62.2,
Rings: 3, SAS: 4.53

QED: 0.57, TPSA: 87.4,
Rings: 1, SAS: 3.09

QED: 0.83, TPSA: 84.4,
Rings: 3, SAS: 3.04

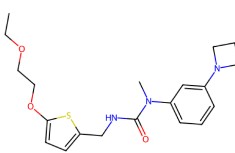

QED: 0.66, TPSA: 54.0,
Rings: 3, SAS: 2.62

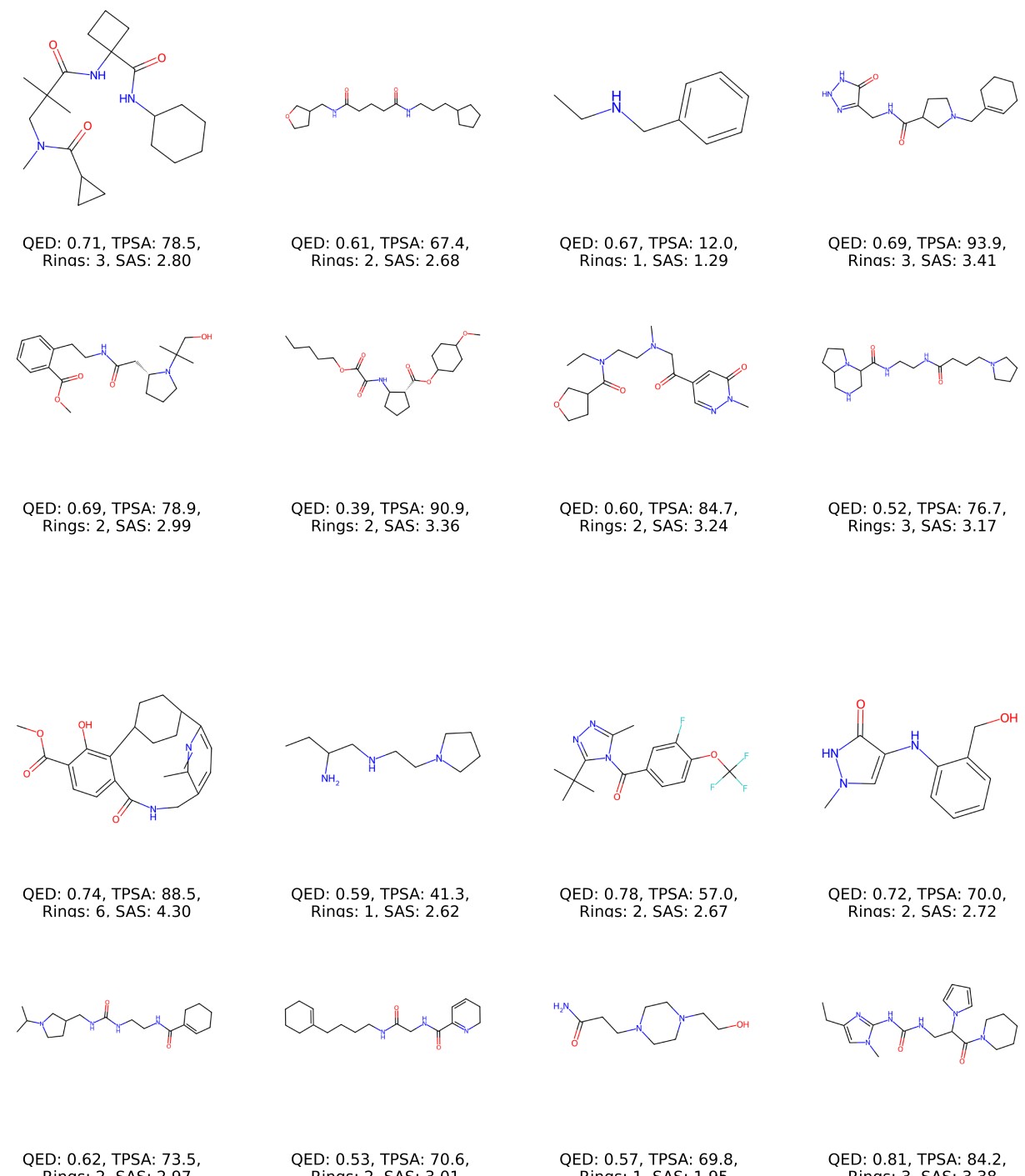

*Figure 3.* Some randomly chosen successful molecules from the pretrained A-GFN.

## D. Conditionals and Rewards

When higher values of a molecular property $p$ or task, within desired range $c_p = (c_{low}, c_{high})$ are preferred, $preference\_direction$ is set to $\vec{d} > 0$ and the corresponding reward is as defined in eq.2. Similarly, when lower values are preferred,

$$R_p(x|c_p, \vec{d} < 0) = \text{reward}(p_x \mid c_p, \vec{d} < 0) = \begin{cases} \exp\left(-\frac{(c_{low} - p_x)}{\lambda}\right) & \text{if } p_x < c_{low} \\ 0.5 * \exp\left(-\frac{(p_x - c_{high})}{\lambda}\right) & \text{if } p_x > c_{high} \\ \frac{-0.5*(p_x - c_{low})}{(c_{high} - c_{low})} + 1 & \text{otherwise} \end{cases} \tag{7}$$

When there is no preference i.e. $\vec{d} = 0$, the reward is defined as

$$R_p(x|c_p, \vec{d} = 0) = \text{reward}(p_x \mid c_p, \vec{d} = 0) = \begin{cases} \exp\left(-\frac{(c_{low} - p_x)}{\lambda}\right) & \text{if } p_x < c_{low} \\ \exp\left(-\frac{(p_x - c_{high})}{\lambda}\right) & \text{if } p_x > c_{high} \\ 1 & \text{otherwise} \end{cases} \tag{8}$$

The rate parameter $\lambda$ is set for each property individually- $\lambda_{QED}=\lambda_{SAS}=\lambda_{NumRings} = 1$, and $\lambda_{TPSA}=20$.

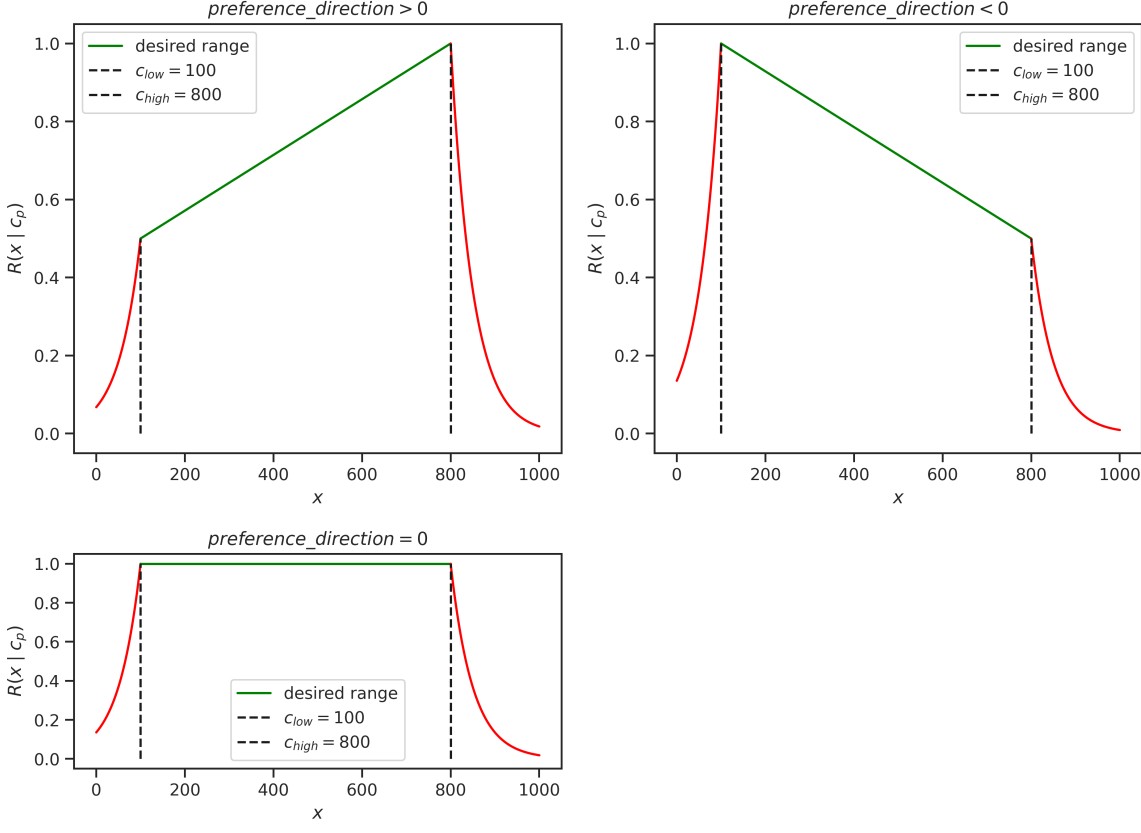

*Figure 4.* Visual representation of reward as a function of $preference\_direction$.

*Table 8.* TPSA Ranges for Different Finetuning Objectives

| Finetuning Objective | TPSA Range |
|---|---|
| Property Optimization | [60,100,0] |
| Preserved Property Constrained Optimization | [60,100,0] |
| DRA Property Constrained Optimization | [100,120,0] / [40,60,0] |

| Task | Finetuning Objective | Task Range |
|---|---|---|
| Mol.Wt. | Property Optimization | [100,800,-1] |
| | Preserved Property Constrained Optimization | [302,800,-1] |
| | DRA Property Constrained Optimization | [340,800,-1] / [300,800,-1] |
| logP | Property Optimization | [-5,6,-1] |
| | Preserved Property Constrained Optimization | [-1.65,5,-1] |
| | DRA Property Constrained Optimization | [1.5,5,-1] / [2.4,5,-1] |
| LD50 | Property Optimization | [2,6,1] |
| | Preserved Property Constrained Optimization | [2.3435,5.541,1] |
| | DRA Property Constrained Optimization | [2.12,4.38,1] / [2.448,5.513,1] |
| Hep. Clearance | Property Optimization | N/A |
| | Preserved Property Constrained Optimization | [3.0,13.18,-1] |
| | DRA Property Constrained Optimization | [3.0,20.0,-1] / [3.0,38.33,-1] |
| Mic. Clearance | Property Optimization | N/A |
| | Preserved Property Constrained Optimization | [5.0,15.0,0] |
| | DRA Property Constrained Optimization | [5.0,15.0,0] / [5.0,15.0,0] |
| PPBR | Property Optimization | N/A |
| | Preserved Property Constrained Optimization | [10.09,95.77,-1] |
| | DRA Property Constrained Optimization | [62.4,95.72,-1] / [11.18,93.94,-1] |
| Caco2 | Property Optimization | N/A |
| | Preserved Property Constrained Optimization | [-4.7545,-4.06,1] |
| | DRA Property Constrained Optimization | [-5.64,-4.95,1] / [-4.597,-4.01,1] |
| Aq.Solv. | Property Optimization | N/A |
| | Preserved Property Constrained Optimization | [-2.798,0.623,1] |
| | DRA Property Constrained Optimization | [-2.71,0.5402,1] / [-3.085,0.8177,1] |
| Lipophil. | Property Optimization | N/A |
| | Preserved Property Constrained Optimization | [-1.45,4.48,0] |
| | DRA Property Constrained Optimization | [-1.2,3.7,0] / [-1.31,4.29,0] |

*Table 9.* Task desired property ranges for different finetuning objectives. The QED, SAS, and Num Rings conditional ranges are [0.65, 0.8, 0], [1,3,0], and [1,3,1] respectively across all experiments where the ordering is $[c_{low}, c_{high}, \vec{d}]$.

# E. Evaluation Metrics

In line with earlier works (Brown et al., 2019; You et al., 2018), we employ following set of metrics to ensure a robust comparison across different generative methods.

**Validity**: A molecule is considered valid if it successfully passes RDKit's sanitization checks. Note that in the proposed method, masking makes this trivially 1.

**Diversity**: For a set of generated molecules, diversity is defined based on the Tanimoto similarity of Morgan Fingerprint representation of molecules.

$$Diversity = 1 - \frac{2}{N(N-1)} \sum_{1 \leq i \leq j \leq N} \frac{|M_i \cap M_j|}{|M_i \cup M_j|}$$

**Uniqueness**: The ratio of distinct canonicalized SMILES strings (without stereochemistry) to the total number of generated molecules, after filtering out duplicates and invalid structures.

**Novelty**: The ratio of unique generated molecules, that are not present in the pretraining ZINC dataset, to the total number of generated molecules.

**Time**: For each task, a fixed time budget is allotted to A-GFN for finetuning and task-training. Other baseline methods are allowed to run for time needed to run their default configuration.

**#Modes**: quantifies the number of distinct, high-reward molecular modes identified by A-GFN model. We define a mode as a molecule whose reward exceeds a threshold, typically set at 0.5. For a molecule to be counted as a new mode, its Tanimoto similarity to any previously identified mode must be less than a specified threshold (typically 0.5), ensuring that only sufficiently diverse molecules are counted.

**RW#C, RW#S, RWTD**: Reward-weighted #circles (RW#C), number of scaffolds (RW#S), and Tanimoto diversity (RWTD) are composite metrics that evaluate molecular diversity while prioritizing high-quality candidates. Each metric is computed as the product of the top-K (typically 100) reward score ($R_{topK}$)) and the respective molecular diversity measure

**Normalized L1-dist**: This measures how far a generated molecule's properties deviate from their target ranges. For each property, the L1 distance is calculated by taking the absolute difference between the generated property value and the $10^{th}$ percentile value in desired range, and then normalized by the respective property range to ensure comparability across properties. This allows us to assess how well the generated molecules meet multi-objective constraints in a unified metric.

**Success Percent**: For a set of $N$ generated graphs $\{G\}_{i=1}^N$ and conditionals $C_{task}$ for given task, we define success percentage as

$$S_{\text{task}} = \frac{1}{N} \sum_{i=1}^{N} \left( \frac{1}{|\mathcal{C}_{\text{task}}|} \sum_{c \in \mathcal{C}_{\text{task}}} \mathbb{I}_c(G_i) \right) \times 100$$

where, $\mathbb{I}_c(G)$ counts the molecules within desired conditional ranges and defined as

$$\mathbb{I}_c(G) = \begin{cases} \mathbb{I}\left(|c(G) - c_{\text{low}}| \leq 0.1 \cdot |c_{\text{low}}|\right), & \text{if } d_p < 0 \\ \mathbb{I}\left(|c(G) - c_{\text{high}}| \leq 0.1 \cdot |c_{\text{high}}|\right), & \text{if } d_p > 0 \\ \mathbb{I}\left(c_{\text{low}} \leq c(G) \leq c_{\text{high}}\right), & \text{if } d_p = 0 \end{cases}$$

where, for any considered task, $c(G)$ is the the value of the conditional property $c$ for the molecular graph $G$, $c_{low}$ and $c_{high}$ are the lower and upper bounds of $c$, $d_p(c)$ indicates whether lower values ($< 0$), higher values ($> 0$), or values within a range ($= 0$) are preferred for that conditional.

**Hit Ratio**: quantifies the percentage of molecules that meet the following criteria: docking score better than the median of known actives, QED > 0.5, and SA < 5 (Guo & Schwaller, 2024b).

**Novel Hit Ratio**: quantifies the percentage of generated molecules that are both unique and novel (Lee et al., 2023b). Novel molecules are defined as those with a maximum Tanimoto similarity below 0.4 to any molecule in the training set. Hits are defined as molecules meeting the following criteria: DS < the median DS of known actives, QED > 0.5, and SA < 5.

# F. Experiments

## F.1. Ablation Studies

### F.1.1. EFFECT OF OFFLINE DATASET SIZE ON PRETRAINING

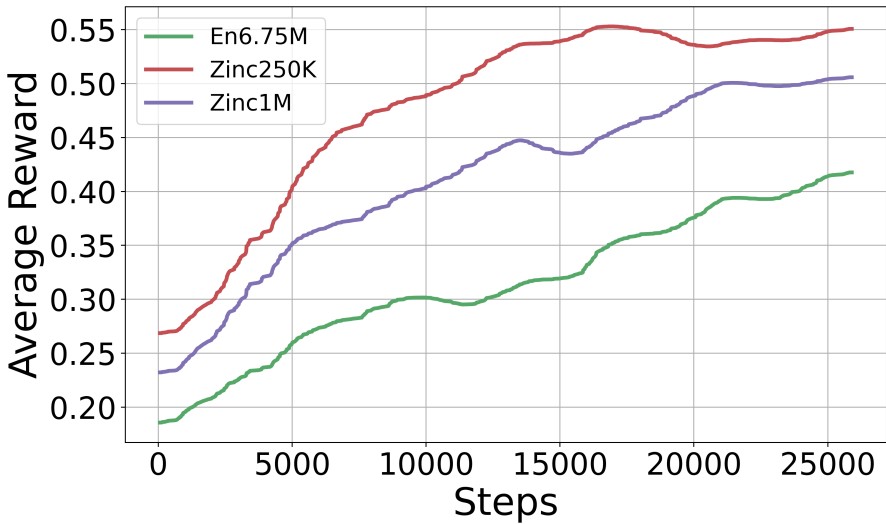

*Figure 5.* Smoothed Reward vs Steps for Drug-like Datasets. Three different offline datasets of varying sizes such as Zinc 250k, Zinc 1M, and Enamine 6.75M samples are considered

Figure 5 above shows how the average reward evolves during pre-training for three different offline datasets: En6.75M, Zinc250K, and Zinc1M. Smaller datasets (Zinc250K) lead to faster initial learning due to reduced diversity, but plateaus quickly as the model converges quickly to a local optimum. On the other hand, larger datasets (En6.75M) require more steps to converge due to higher diversity, but ultimately achieve better generalization, as seen in the final rewards.

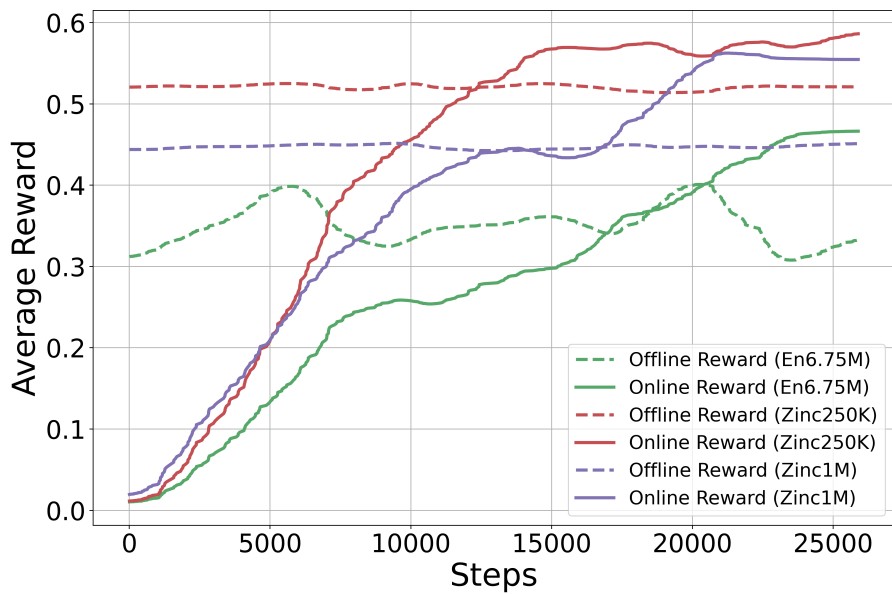

*Figure 6.* Offline and Online Rewards vs Steps for Different Datasets. Three different datasets of varying sizes such as Zinc 250k, Zinc 1M, and Enamine 6.75M samples are considered

Figure 6 shows the average rewards for offline and online pretraining across different dataset sizes. Offline rewards plateau for all datasets, indicating that offline pretraining alone cannot fully explore and works best in combination with the adaptability of online training. Larger datasets like En6.75M enable higher final rewards, which showcases their ability to avoid overfitting.

F.1.2. EFFECT OF TEMPERATURE ($\beta$)

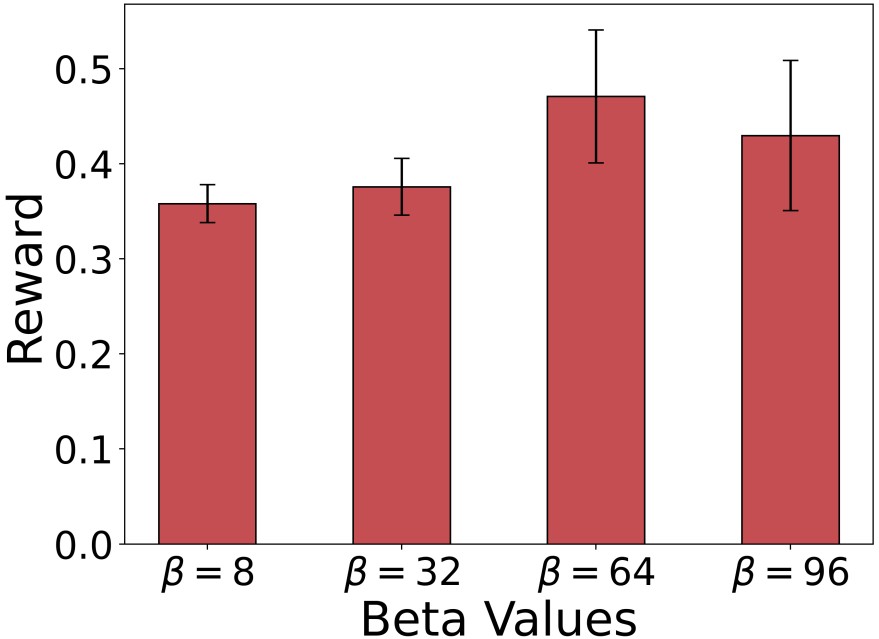

*Figure 7.* Effect of Temperature Parameter $\beta$ on Reward Scale

Figure 7 shows the influence of temperature $\beta$ on reward performance, highlighting the trade-off between exploration and exploitation in GFlowNet. Lower values of $\beta$ encourage greater exploration, allowing the model to traverse state space more freely. However, given the vast molecular space, this setting may significantly slow down convergence, as it requires extensive searching before identifying high-reward molecular modes. Higher $\beta$ values result in a diversity-speed trade-off as they promote exploitation, enabling the model to focus on high-reward molecules it encounters early in the training. While this improves convergence speed, it risks early convergence by overexploiting specific regions of the manifold and failing to discover all possible high-reward modes. Excessively high $\beta$ values also introduce instability in learning.

F.1.3. EFFECT OF OFFLINE DATA ON PRETRAINING

In Figure 8 we compare the average online reward achieved by an A-GFN model under three different pretraining schemes: Purely Online, Purely Offline and Online + Offline. Models with offline pretraining (Purely Offline and Online + Offline) achieve higher rewards compared to the purely online scheme. This shows that offline pretraining helps the model learn a good initial representation, reducing the burden on online training. Moreover, the plateau in the Purely Offline curve suggests that while offline data provides a good starting point, it lacks the adaptability and exploration benefits provided by online learning. Purely online training collapses under the weight of large state space.

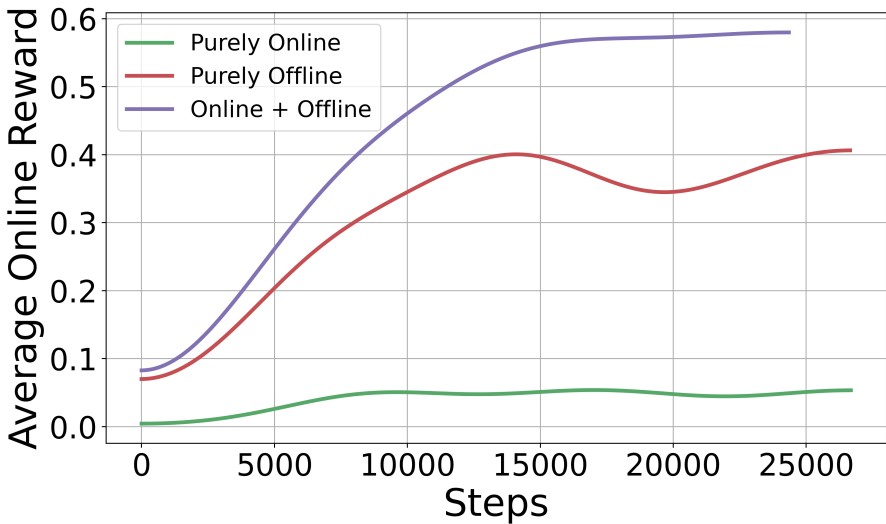

*Figure 8.* Rewards Vs Steps for A-GFN Sampled Trajectories for Different Pretraining Configurations - Online, Offline and Online + Offline

### F.1.4. EFFECT OF PARAMETERS SCALING

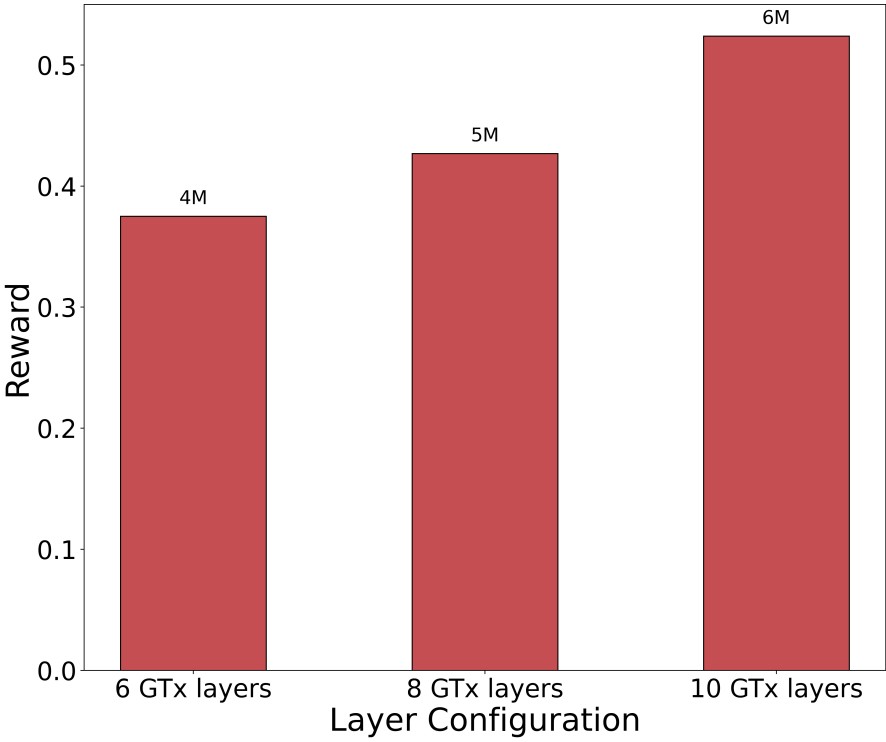

*Figure 9.* Number of A-GFN Parameters Vs Rewards. Number of parameters written over the bars

Figure 9 demonstrates the benefit of parameter scaling for A-GFN, where increasing the number of Graph Transformer layers leads to higher average rewards. This aligns with the intuition that deeper models provide better representations and this improvement in reward suggests that the GFlowNet becomes more efficient at sampling diverse, high-reward trajectories with the increase in number of parameters

## F.2. Property Optimization

| Task | Method | Chem. Space Coverage | | Diversity | | | | | | Reward | | |
|---|---|---|---|---|---|---|---|---|---|---|---|---|
| | | #Modes | RW#C | RW#S | RWTD | L1-dist ↓ | Novelty | Uniqueness | Success % | Top-1 | Top-10 | Top-100 |
| Mol.Wt. | AGFN Task Trained | 0 | 20.14 | 2.43 | 0.12 | 4.5624 | 1 | 1 | 0 | 0.0009 | 0.0008 | 1.00E-05 |
| | **AGFN Finetuned** | **992** | **79.26** | **692.97** | **0.91** | **0.0603** | **1** | **1** | **1.7** | **0.9999** | **0.9956** | **9.80E-01** |
| | AGFN Finetuned w/ RTB | 489 | 118.12 | 240.25 | 0.83 | 0.0803 | 0.835 | 0.835 | 0.1 | 0.9841 | 0.982 | 9.67E-01 |
| logP | AGFN Task Trained | 0 | 11.54 | 5.24 | 0.16 | 2.6541 | 1 | 1 | 0 | 0.0009 | 0.0009 | 1.00E-05 |
| | **AGFN Finetuned** | **994** | **439.91** | **116.61** | **0.84** | **0.0855** | **1** | **1** | **28.1** | **0.9996** | **0.9983** | **9.86E-01** |
| | AGFN Finetuned w/ RTB | 977 | 185.89 | 936.42 | 0.82 | 0.1536 | 0.998 | 0.998 | 7.9 | 0.9963 | 0.9914 | 9.64E-01 |
| Toxicity (LD50) | AGFN Task Trained | 0 | 6.84 | 3.94 | 0.02 | 1.5617 | 1 | 1 | 0 | 0.0008 | 0.0008 | 1.00E-05 |
| | **AGFN Finetuned** | **0** | **154.46** | **6.91** | **0.16** | **0.7504** | **1** | **1** | **0** | **0.2688** | **0.2364** | **1.87E-01** |
| | AGFN Finetuned w/ RTB | 0 | 43.4 | 108.90 | 0.09 | 0.8574 | 1 | 1 | 0 | 0.1525 | 0.1383 | 1.09E-01 |

*Table 10.* Comparing the effectiveness of finetuned atomic-GFlowNet over atomic-GFlowNet trained from scratch for 3 tasks with property optimized objective.

## F.3. Property Targeting

| DRA | Method | Chem. Space Coverage | | Diversity | | | | | MOO metrics | | | | | Reward | | |
|---|---|---|---|---|---|---|---|---|---|---|---|---|---|---|---|---|
| | | | | | | | | | | | L1-dist ↓ | | | | | |
| | | #Modes | RW# | RW#S | RWTD | Nov. | Uniq. | Success % | HV | TPSA | #Rings | SAS | QED | Top-1 | Top-10 | Top-100 |
| Low TPSA | AGFN Tasktrained | 0 | 0.037 | 0.037 | 0 | 0.02 | 0.02 | 0 | 0.0373 | 0 | 1.5 | 0.62 | 2.40 | 0.03 | 0.03 | 0.03 |
| | AGFN Finetuned | 0 | 0.0096 | 0.0064 | 0.00007 | 0.037 | 0.037 | 0 | 0.1228 | 16.243 | 1.5 | 0.7209 | 4.0357 | 0.1196 | 0.0318 | 0.0032 |
| | **AGFN FT w/ RTB** | **79** | **12** | **62** | **0.567** | **0.799** | **0.8** | **94.6** | **1** | **0.025** | **0.028** | **0.0036** | **0.0371** | **1** | **1** | **1** |
| High TPSA | AGFN Tasktrained | 0 | 0.447 | 0.298 | 0.0773 | 0.240 | 0.240 | 0 | 0.1689 | 0.002 | 1.5 | 0.004 | 1.35 | 0.163 | 0.163 | 0.149 |
| | AGFN Finetuned | 0 | 0.0252 | 0.0168 | 0.0092 | 0.006 | 0.006 | 0 | 0.0144 | 2.5189 | 1.5 | 0.1146 | 2.3395 | 0.0109 | 0.0086 | 0.0084 |
| | **AGFN FT w/ RTB** | **147** | **18** | **225** | **0.682** | **0.653** | **0.653** | **94.5** | **0.9999** | **0.0456** | **0.03** | **0.0099** | **0.0518** | **1** | **1** | **1** |

*Table 11.* Comparing the effectiveness of finetuned A-GFN over A-GFN trained from scratch for TPSA property targeting objective.

## F.4. Property Constrained Optimization

We present the results of extensive benchmarking across relevant drug discovery tasks. These tasks encompass all regression benchmarks included in the Therapeutics Data Commons (TDC) ADME-T group. The benchmark rewards were generated using a gradient-boosted decision tree - CatBoost (Notwell & Wood, 2023) trained on the respective datasets provided by TDC. In addition to these, we evaluated two tasks with deterministic rewards computed using RDKit: molecular weight and logP. For each property-constrained optimization variant—Preserved and DRA (low TPSA and high TPSA)—the desired ranges for each task are summarized in Table 9.

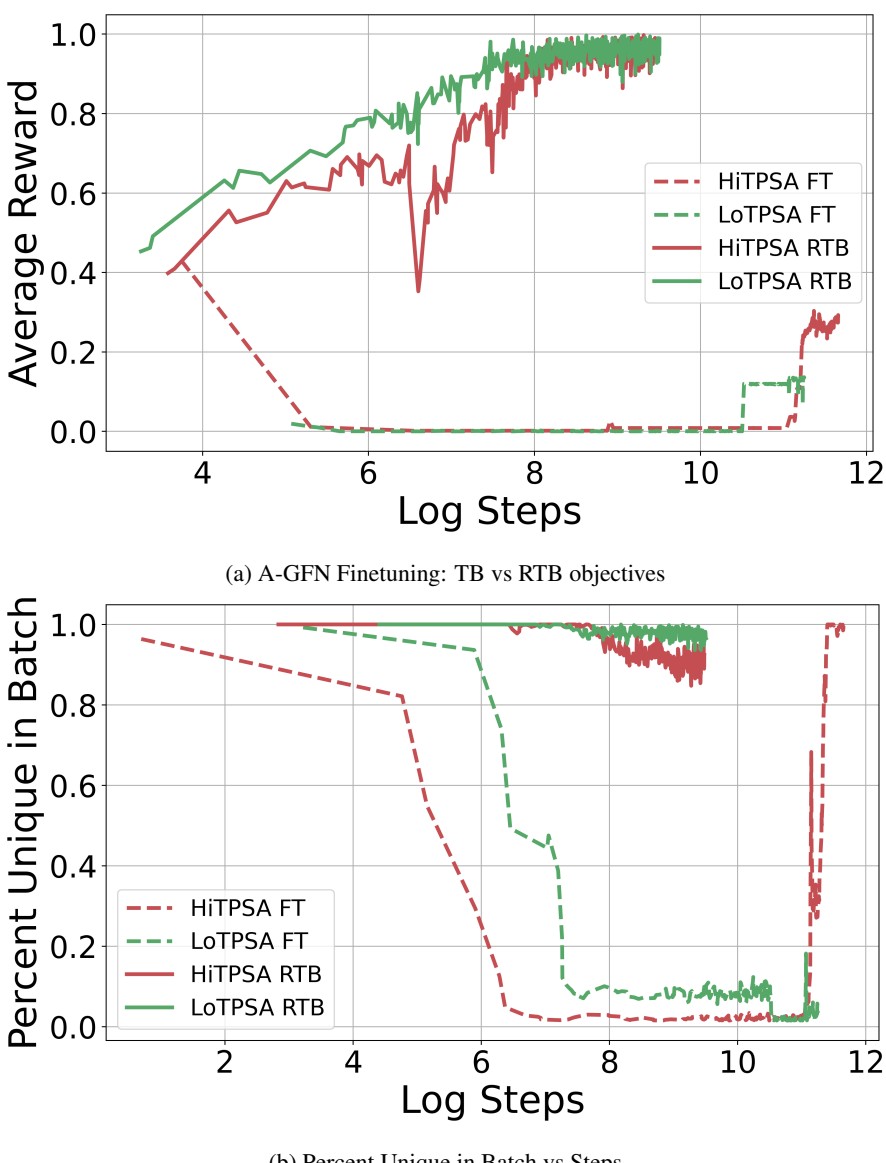

(a) A-GFN Finetuning: TB vs RTB objectives

(b) Percent Unique in Batch vs Steps

*Figure 10.* AGFN Finetuning with and without the RTB objective and the effect of overtraining

### F.4.1. TASK : MOLECULAR WEIGHT

| | Method | Chem. Space Coverage | | | | Diversity | | | MOO metrics | | | | | | Reward | | |
|---|---|---|---|---|---|---|---|---|---|---|---|---|---|---|---|---|---|
| | | | | | | | | | | | L1-dist ↓ | | | | | | |
| | | #modes | RW#C | RW#S | RWTD | Nov | Uniq | Success % | HV | TPSA | #Rings | SAS | QED | Task | Top-1 | Top-10 | Top-100 |
| Preserved | MolDQN | 0 | 4.992 | 5.568 | 0.014 | 1 | 1 | 0.0 | 0.070 | 0.006 | 0.79 | 1.495 | 2.243 | 0.051 | 0.03 | 0.026 | 0.016 |
| | Reinvent | 94 | 4.686 | 368.632 | 0.532 | 0.998 | 1 | 23.5 | 0.861 | 0 | 0 | 0 | 0.237 | 0.083 | 0.819 | 0.808 | 0.781 |
| | Frag GFlowNet | 24 | 76.725 | 322.710 | 0.396 | 1 | 1 | 22.0 | 0.751 | 0.000 | 0.035 | 0.006 | 0.144 | 0.094 | 0.680 | 0.564 | 0.465 |
| | Graph MCTS | 1 | 100.480 | 179.294 | 0.276 | 1 | 1 | 1.6 | 0.645 | 0 | 0.371 | 0.357 | 0.508 | 0.06 | 0.53 | 0.451 | 0.314 |
| | AGFN Tasktrained | 0 | 0.001 | 0.000 | 0.000 | 1 | 1 | 0.1 | 0.109 | 5.792 | 1.500 | 1.862 | 2.493 | 0.153 | <10^-4 | <10^-4 | <10^-4 |
| | AGFN Finetuned | 0 | 0.000 | 0.000 | 0.000 | 0 | 0 | 0.0 | 0.000 | 6.279 | 1.500 | 1.180 | 2.468 | 0.000 | 0.000 | 0.000 | 0.000 |
| | AGFN FT w/ data | 532 | 314.885 | 824.023 | **0.766** | 1 | 1 | 2.0 | 0.986 | 0.033 | 0.498 | 0.174 | 0.279 | 0.000 | 0.969 | 0.943 | 0.887 |
| | AGFN FT w/ RTB | 675 | 52.494 | **903.654** | 0.756 | 0.997 | 0.997 | 6.9 | 0.977 | 0.022 | 0.104 | 0.019 | 0.266 | 0.109 | 0.973 | 0.964 | **0.937** |
| | AGFN FT w/ RTB + data | 466 | **349.040** | 846.422 | 0.755 | 1 | 1 | 0.3 | **0.990** | 0.025 | 0.398 | 0.259 | 0.26 | 0.136 | 0.957 | 0.924 | 0.873 |
| Low TPSA | MolDQN | 0 | 0.560 | 0.662 | 0.002 | 1 | 1 | 0.0 | 0.027 | 1.028 | 0.806 | 1.432 | 2.027 | 0.044 | 0.017 | 0.011 | 0.002 |
| | Reinvent | 204 | 19.775 | 253.120 | 0.589 | 0.998 | 1 | 37.2 | 0.883 | 0 | 0.038 | 0 | 0.091 | 0.069 | 0.847 | 0.832 | 0.791 |
| | Frag GFlowNet | 38 | 88.528 | 482.377 | 0.442 | 1 | 1 | 11.2 | 0.811 | 0.001 | 0.174 | 0.216 | 0.279 | 0.073 | 0.785 | 0.682 | 0.503 |
| | Graph MCTS | 3 | 128.734 | 215.343 | 0.300 | 1 | 1 | 2.0 | 0.690 | 0.002 | 0.419 | 0.49 | 0.388 | 0.064 | 0.625 | 0.477 | 0.337 |
| | AGFN Tasktrained | 0 | 0.001 | 0.056 | 0.000 | 1 | 1 | 0.0 | 0.002 | 1.623 | 1.500 | 2.065 | 0.995 | 0.234 | <10^-4 | <10^-4 | <10^-4 |
| | AGFN Finetuned | 0 | 0.060 | 0.160 | 0.001 | 0 | 0 | 0.0 | 0.106 | 1.994 | 1.497 | 0.002 | 2.913 | 0.000 | 0.081 | 0.059 | 0.020 |
| | AGFN FT w/ data | 449 | **298.034** | **846.906** | **0.751** | 1 | 1 | 0.8 | 0.929 | 0.053 | 0.449 | 0.251 | 0.273 | 0.000 | 0.963 | 0.931 | 0.874 |
| | AGFN FT w/ RTB | 786 | 80.070 | **897.726** | **0.772** | 0.994 | 0.994 | 10.0 | **0.990** | 0.072 | 0.061 | 0.036 | 0.260 | 0.092 | 0.973 | 0.967 | **0.942** |
| | AGFN FT w/ RTB + data | 505 | 249.893 | 839.296 | 0.735 | 0.999 | 0.999 | 0.0 | 0.9457 | 0.072 | 0.363 | 0.184 | 0.21 | 0.152 | 0.925 | 0.899 | 0.862 |
| High TPSA | MolDQN | 0 | 1.944 | 2.034 | 0.005 | 1 | 1 | 0.0 | 0.029 | 0.37 | 0.843 | 1.513 | 2.978 | 0.039 | 0.015 | 0.012 | 0.006 |
| | Reinvent | 129 | 11.628 | 269.496 | 0.503 | 1 | 1 | 37.3 | 0.758 | 0 | 0.026 | 0 | 0.107 | 0.088 | 0.704 | 0.699 | 0.684 |
| | Frag GFlowNet | 1 | 61.500 | 235.500 | 0.219 | 1 | 1 | 2.4 | 0.592 | 0.002 | 0.217 | 0.394 | 0.856 | 0.119 | 0.51 | 0.412 | 0.25 |
| | Graph MCTS | 0 | 44.451 | 61.776 | 0.088 | 1 | 1 | 0.0 | 0.444 | 0.005 | 0.579 | 0.938 | 1.871 | 0.074 | 0.256 | 0.213 | 0.099 |
| | AGFN Tasktrained | 0 | 0.000 | 0.002 | 0.000 | 1 | 1 | 0.0 | 0.001 | 2.653 | 1.500 | 1.620 | 1.423 | 0.365 | <10^-4 | <10^-4 | <10^-4 |
| | AGFN Finetuned | 0 | 0.000 | 0.000 | 0.000 | 0 | 0 | 0.0 | 0.001 | 4.999 | 1.500 | 0.003 | 3.829 | 0.477 | 0.001 | 0.001 | 0.000 |
| | AGFN FT w/ data | 516 | **334.683** | 870.720 | **0.785** | 1 | 1 | 8.5 | **0.998** | 0.204 | 0.400 | 0.188 | 0.227 | 0.000 | 0.975 | 0.953 | 0.907 |
| | AGFN FT w/ RTB | 672 | 110.003 | **911.054** | 0.778 | 0.992 | 0.992 | 21.1 | 0.985 | 0.165 | 0.123 | 0.049 | 0.567 | 0.096 | 0.982 | 0.980 | **0.940** |
| | AGFN FT w/ RTB + data | 467 | **328.742** | 830.319 | 0.771 | 1 | 1 | 5.5 | 0.946 | 0.234 | 0.364 | 0.18 | 0.294 | 0.122 | 0.997 | 0.963 | 0.891 |

*Table 12.* Comparing the effectiveness of finetuned A-GFN over A-GFN trained from scratch and other baselines for property-constrained optimization objective for Mol.Wt. task.

F.4.2. TASK: LOGP

| | Method | Chem. Space Coverage | | | | Diversity | | | MOO metrics | L1-dist ↓ | | | | | Reward | | |
|---|---|---|---|---|---|---|---|---|---|---|---|---|---|---|---|---|---|
| | | #modes | RW#C | RW#S | RWTD | Nov | Uniq | Success % | HV | TPSA | #Rings | SAS | QED | Task | Top-1 | Top-10 | Top-100 |
| **Preserved** | MolDQN | 0 | 5.729 | 30.330 | 0.255 | 1 | 1 | 2.1 | 0.582 | 0 | 0.353 | 0.14 | 0.096 | 0.175 | 0.46 | 0.431 | 0.337 |
| | Reinvent | 139 | 7.590 | 204.171 | 0.545 | 0.997 | 1 | 30.8 | 0.835 | 0 | 0 | 0 | 0.233 | 0.106 | 0.798 | 0.787 | 0.759 |
| | Frag GFlowNet | 51 | 46.096 | 366.088 | 0.461 | 1 | 1 | 11.3 | 0.820 | 0 | 0.13 | 0.056 | 0.204 | 0.135 | 0.711 | 0.632 | 0.536 |
| | Graph MCTS | 3 | 66.256 | 140.592 | 0.347 | 1 | 1 | 0.2 | 0.713 | 0 | 0.524 | 0.055 | 0.17 | 0.233 | 0.523 | 0.493 | 0.404 |
| | AGFN Tasktrained | 0 | 0.000 | 0.004 | 0.000 | 1 | 1 | 0.0 | 0.0001 | 0.065 | 0.956 | 2.862 | 1.000 | 0.446 | 1.00E-04 | 1.00E-04 | <10^-4 |
| | AGFN Finetuned | 1 | 43.703 | 42.468 | 0.040 | 1 | 1 | 0.0 | 0.677 | 0.086 | 0.827 | 2.047 | 3.622 | 0.311 | 0.6267 | 0.2412 | 0.0441 |
| | AGFN FT w/ data | 419 | **298.290** | 809.401 | 0.735 | 1 | 1 | 4.0 | 0.991 | 0.031 | 0.417 | 0.172 | 0.266 | 0.187 | 0.9995 | 0.9779 | 0.8547 |
| | AGFN FT w/ RTB | 708 | 71.569 | **948.047** | **0.793** | 0.997 | 0.997 | 24.6 | **0.996** | 0.011 | 0.069 | 0.029 | 0.174 | 0.126 | 0.9989 | 0.9981 | **0.9804** |
| | AGFN FT w/ RTB + data | 351 | 281.655 | 810.801 | 0.716 | 1 | 1 | 2.2 | 0.983 | 0.026 | 0.364 | 0.191 | 0.266 | 0.259 | 0.9833 | 0.9575 | 0.8333 |
| **Low TPSA** | MolDQN | 0 | 6.344 | 15.860 | 0.189 | 1 | 1 | 0 | 0.517 | 0 | 0.514 | 0.292 | 0.106 | 0.301 | 0.302 | 0.287 | 0.244 |
| | Reinvent | 162 | 11.640 | 172.272 | 0.578 | 0.996 | 1 | 36.3 | 0.889 | 0 | 0.032 | 0 | 0.237 | 0.141 | 0.837 | 0.82 | 0.776 |
| | Frag GFlowNet | 106 | 24.880 | 499.466 | 0.524 | 1 | 1 | 27.3 | 0.88 | 0 | 0.093 | 0.005 | 0.168 | 0.169 | 0.741 | 0.712 | 0.622 |
| | Graph MCTS | 13 | 75.240 | 164.160 | 0.392 | 1 | 1 | 0.2 | 0.79 | 0 | 0.493 | 0.036 | 0.16 | 0.234 | 0.592 | 0.536 | 0.456 |
| | AGFN Tasktrained | 0 | 0.001 | 0.001 | 0.000 | 1 | 1 | 0.0 | 0.0001 | 0.351 | 1.235 | 0.321 | 0.654 | 1.235 | 0.003 | 0.002 | 0.000 |
| | AGFN Finetuned | 0 | 29.282 | 28.077 | 0.027 | 1 | 1 | 0.0 | 0.534 | 0.181 | 0.961 | 1.996 | 2.918 | 0.304 | 0.395 | 0.146 | 0.029 |
| | AGFN FT w/ data | 317 | **258.635** | 763.172 | 0.684 | 1 | 1 | 3.1 | 0.995 | 0.080 | 0.432 | 0.182 | 0.278 | 0.250 | 0.983 | 0.937 | 0.796 |
| | AGFN FT w/ RTB | 479 | 55.395 | **901.344** | **0.753** | 0.992 | 0.992 | 15.2 | **0.997** | 0.079 | 0.154 | 0.039 | 0.182 | 0.215 | 0.990 | 0.984 | **0.939** |
| | AGFN FT w/ RTB + data | 112 | 251.667 | 633.039 | 0.556 | 0.999 | 0.999 | 1.8 | 0.986 | 0.145 | 0.331 | 0.288 | 0.358 | 0.485 | 0.970 | 0.917 | 0.645 |
| **High TPSA** | MolDQN | 0 | 6.383 | 5.447 | 0.012 | 1 | 1 | 0 | 0.116 | 0 | 0.901 | 1.098 | 2.114 | 0.252 | 0.053 | 0.03 | 0.013 |
| | Reinvent | 102 | 6.066 | 324.194 | 0.463 | 1 | 1 | 33 | 0.748 | 0 | 0.007 | 0 | 0.131 | 0.091 | 0.729 | 0.699 | 0.674 |
| | Frag GFlowNet | 0 | 47.775 | 224.133 | 0.237 | 1 | 1 | 1.3 | 0.652 | 0 | 0.22 | 0.235 | 0.46 | 0.206 | 0.483 | 0.419 | 0.273 |
| | Graph MCTS | 0 | 50.841 | 85.617 | 0.165 | 1 | 1 | 0 | 0.550 | 0 | 0.554 | 0.286 | 1.011 | 0.176 | 0.349 | 0.264 | 0.189 |
| | AGFN Tasktrained | 0 | 0.000 | 0.001 | 0.000 | 1 | 1 | 0.0 | 0.002 | 1.360 | 1.654 | 0.032 | 0.900 | 1.220 | 0.001 | 0.001 | <10^-4 |
| | AGFN Finetuned | 0 | 33.998 | 32.668 | 0.031 | 1 | 1 | 0.0 | 0.454 | 0.288 | 0.911 | 2.122 | 3.941 | 0.275 | 0.385 | 0.198 | 3.41E-02 |
| | AGFN FT w/ data | 115 | 233.989 | 629.417 | 0.564 | 1 | 1 | 1.6 | **0.998** | 0.787 | 0.419 | 0.190 | 0.279 | 0.173 | 0.953 | 0.883 | 0.654 |
| | AGFN FT w/ RTB | 490 | 55.200 | **898.176** | **0.741** | 0.991 | 0.991 | 7.6 | 0.994 | 0.137 | 0.206 | 0.031 | 0.389 | 0.153 | 0.998 | 0.990 | **0.936** |
| | AGFN FT w/ RTB + data | 260 | **256.113** | 744.280 | 0.667 | 1 | 1 | 2.8 | 0.994 | 0.468 | 0.355 | 0.173 | 0.316 | 0.202 | 0.995 | 0.941 | 0.776 |

*Table 13.* Comparing the effectiveness of finetuned A-GFN over A-GFN trained from scratch and other baselines for property-constrained optimization objective for logP task.

F.4.3. TASK: ABSORPTION - CELL EFFECTIVE PERMEABILITY

| | Method | Chem. Space Coverage | | Diversity | | | | | MOO metrics | | | | | | Reward | | |
|---|---|---|---|---|---|---|---|---|---|---|---|---|---|---|---|---|---|
| | | | | | | | | | | L1-dist ↓ | | | | | | | |
| | | #modes | RW#C | RW#S | RWTD | Nov | Uniq | Success % | HV | TPSA | #Rings | SAS | QED | Task | Top-1 | Top-10 | Top-100 |
| Preserved | MolDQN | 0 | 0.316 | 0.272 | 0.001 | 1 | 1 | 0.0 | 0.002 | 0.000 | 0.919 | 0.500 | 1.167 | 0.649 | 0.003 | 0.001 | 0.001 |
| | Reinvent | 0 | 1.079 | 5.785 | 0.005 | 1 | 1 | 43.3 | 0.004 | 0.000 | 0.230 | 0.004 | 0.207 | 0.243 | 0.008 | 0.007 | 0.006 |
| | Frag GFlowNet | 0 | 0.435 | 4.241 | 0.005 | 1 | 1 | 49.6 | 0.004 | 0.000 | 0.171 | 0.020 | 0.178 | 0.263 | 0.007 | 0.006 | 0.006 |
| | Graph MCTS | 0 | 0.735 | 1.584 | 0.004 | 1 | 1 | 4.7 | 0.004 | 0.000 | 0.465 | 0.047 | 0.149 | 0.380 | 0.006 | 0.005 | 0.004 |
| | AGFN Tasktrained | 0 | 0.050 | 0.100 | 0.001 | 1 | 1 | 0.0 | 0.005 | 0.200 | 0.500 | 0.400 | 2.000 | 1.500 | 0.004 | 0.002 | 0.001 |
| | AGFN Finetuned | 0 | 0.786 | 0.553 | 0.001 | 1 | 1 | 0.0 | 0.012 | 0.069 | 1.155 | 1.667 | 3.822 | 1.708 | 0.009 | 0.003 | 0.001 |
| | AGFN FT w/ data | 0 | 4.822 | 13.054 | 0.015 | 1 | 1 | 2.2 | 0.035 | 0.045 | 0.350 | 0.131 | 0.372 | 1.149 | 0.030 | 0.026 | 0.017 |
| | AGFN FT w/ RTB | 0 | 56.242 | **436.106** | **0.376** | 1 | 1 | 0.0 | 0.501 | 0.010 | 0.116 | 0.042 | 0.316 | 1.351 | 0.499 | 0.488 | **0.461** |
| | AGFN FT w/ RTB + data | 0 | **174.124** | 381.866 | **0.376** | 0 | 1 | 1.0 | **0.578** | 0.048 | 0.340 | 0.141 | 0.461 | 1.204 | 0.494 | 0.488 | 0.431 |
| Low TPSA | MolDQN | 0 | 0.286 | 0.243 | 0.001 | 1 | 1 | 0.0 | 0.002 | 0.000 | 1.074 | 0.441 | 1.110 | 0.449 | 0.002 | 0.001 | 0.001 |
| | Reinvent | 0 | 0.988 | 5.649 | 0.006 | 1 | 1 | 43.7 | 0.004 | 0.000 | 0.229 | 0.002 | 0.305 | 0.198 | 0.007 | 0.007 | 0.007 |
| | Frag GFlowNet | 0 | 0.556 | 3.931 | 0.005 | 1 | 1 | 32.2 | 0.004 | 0.001 | 0.267 | 0.058 | 0.184 | 0.270 | 0.007 | 0.006 | 0.005 |
| | Graph MCTS | 0 | 0.648 | 1.607 | 0.004 | 1 | 1 | 7.0 | 0.004 | 0.000 | 0.471 | 0.020 | 0.154 | 0.262 | 0.006 | 0.006 | 0.005 |
| | AGFN Tasktrained | 0 | 0.080 | 0.220 | 0.001 | 1 | 1 | 45.0 | 0.006 | 0.220 | 0.550 | 0.420 | 2.200 | 1.600 | 0.005 | 0.003 | 0.002 |
| | AGFN Finetuned | 0 | 1.229 | 0.761 | 0.001 | 1 | 1 | 0.0 | 0.012 | 0.161 | 1.232 | 1.093 | 3.527 | 1.706 | 0.005 | 0.003 | 0.002 |
| | AGFN FT w/ data | 0 | 3.596 | 11.280 | 0.012 | 1 | 1 | 0.0 | 0.027 | 0.130 | 0.301 | 0.122 | 0.359 | 1.286 | 0.020 | 0.018 | 0.014 |
| | AGFN FT w/ RTB | 0 | 26.970 | **423.150** | 0.359 | 1 | 1 | 0.0 | 0.510 | 0.070 | 0.092 | 0.019 | 0.670 | 1.227 | 0.500 | 0.492 | **0.465** |
| | AGFN FT w/ RTB + data | 0 | **143.856** | 421.356 | **0.385** | 1 | 1 | 0.1 | **0.512** | 0.134 | 0.275 | 0.140 | 0.532 | 1.256 | 0.495 | 0.487 | 0.444 |
| High TPSA | MolDQN | 0 | 0.106 | 0.097 | 0.000 | 1 | 1 | 0.0 | 0.000 | 0.000 | 0.829 | 0.848 | 1.902 | 0.342 | 0.000 | 0.000 | 0.000 |
| | Reinvent | 0 | 0.427 | 1.115 | 0.001 | 1 | 1 | 3.3 | 0.002 | 0.000 | 0.392 | 0.175 | 0.525 | 0.829 | 0.002 | 0.002 | 0.001 |
| | Frag GFlowNet | 0 | 0.163 | 1.144 | 0.001 | 1 | 1 | 12.9 | 0.001 | 0.001 | 0.186 | 0.118 | 0.314 | 0.668 | 0.003 | 0.002 | 0.002 |
| | Graph MCTS | 0 | 0.226 | 0.428 | 0.001 | 1 | 1 | 0.7 | 0.001 | 0.001 | 0.638 | 0.181 | 0.964 | 0.348 | 0.002 | 0.002 | 0.001 |
| | AGFN Tasktrained | 0 | 0.090 | 0.360 | 0.002 | 1 | 1 | 40.0 | 0.007 | 0.250 | 0.600 | 0.450 | 2.400 | 1.800 | 0.006 | 0.004 | 0.003 |
| | AGFN Finetuned | 0 | 0.299 | 0.264 | 0.000 | 1 | 1 | 0.0 | 0.004 | 0.309 | 1.017 | 1.962 | 3.869 | 1.011 | 0.002 | 0.001 | 0.000 |
| | AGFN FT w/ data | 0 | 0.816 | 4.243 | 0.007 | 1 | 1 | 14.0 | 0.023 | 0.351 | 0.400 | 0.134 | 0.368 | 0.525 | 0.014 | 0.011 | 0.010 |
| | AGFN FT w/ RTB | 649 | 70.956 | **936.036** | **0.786** | 1 | 1 | 62.2 | **0.997** | 0.116 | 0.079 | 0.039 | 0.316 | 0.191 | 1.000 | 0.998 | **0.972** |
| | AGFN FT w/ RTB + data | 142 | **111.313** | 586.343 | 0.595 | 1 | 1 | 10.3 | 0.993 | 0.447 | 0.315 | 0.129 | 0.498 | 0.384 | 0.960 | 0.915 | 0.709 |

*Table 14.* Comparing the effectiveness of finetuned A-GFN over A-GFN trained from scratch and other baselines for property-constrained optimization objective for Cell Effective Permeability (Caco-2) task.

### F.4.4. TASK: ABSORPTION - LIPOPHILICITY

| | Method | Chem. Space Coverage | | Diversity | | Nov | Uniq | Success % | MOO metrics / L1-dist ↓ | | | | | | Reward | | |
|---|---|---|---|---|---|---|---|---|---|---|---|---|---|---|---|---|---|
| | | #modes | RW#C | RW#S | RWTD | | | | HV | TPSA | #Rings | SAS | QED | Task | Top-1 | Top-10 | Top-100 |
| Preserved | MolDQN | 0 | 1.720 | 14.190 | 0.133 | 1 | 1 | 0.0 | 0.423 | 0.000 | 0.332 | 0.248 | 0.134 | 0.646 | 0.302 | 0.263 | 0.215 |
| | Reinvent | 119 | 6.050 | 277.750 | 0.403 | 0.998 | 1 | 0.0 | 0.723 | 0.000 | 0 | 0 | 0.058 | 0.707 | 0.569 | 0.563 | 0.55 |
| | Frag GFlowNet | 1 | 15.200 | 302.800 | 0.335 | 1 | 1 | 0.1 | 0.710 | 0.00003 | 0.112 | 0.013 | 0.156 | 0.401 | 0.518 | 0.464 | 0.4 |
| | Graph MCTS | 0 | 46.893 | 105.270 | 0.273 | 1 | 1 | 0.0 | 0.639 | 0.000 | 0.504 | 0.047 | 0.149 | 0.509 | 0.378 | 0.366 | 0.319 |
| | AGFN Tasktrained | 0 | 0.399 | 0.238 | 0.006 | 1 | 1 | 0.0 | 0.058 | 0.252 | 1.390 | 1.362 | 3.014 | 0.000 | 0.021 | 0.016 | 0.007 |
| | AGFN Finetuned | 0 | 22.487 | 17.153 | 0.021 | 1 | 1 | 0.0 | 0.263 | 0.082 | 1.163 | 1.948 | 3.929 | 0.000 | 0.178 | 0.086 | 0.023 |
| | AGFN FT w/ data | 616 | **372.000** | 886.00 | **0.879** | 1 | 1 | 32.8 | **1.000** | 0.064 | 0.307 | 0.124 | 0.411 | 0.000 | 1.000 | 1.000 | **1.000** |
| | AGFN FT w/ RTB | 821 | 277.000 | **986.00** | 0.862 | 1 | 1 | 48.7 | **1.000** | 0.026 | 0.199 | 0.074 | 0.273 | 0.000 | 1.000 | 1.000 | **1.000** |
| | AGFN FT w/ RTB + data | 677 | 336.000 | 861.00 | **0.879** | 1 | 1 | 46.2 | **1.000** | 0.056 | 0.306 | 0.103 | 0.310 | 0.000 | 1.000 | 1.000 | **1.000** |
| Low TPSA | MolDQN | 0 | 14.490 | 16.020 | 0.080 | 1 | 1 | 0.0 | 0.353 | 0.001 | 1.085 | 0.193 | 0.898 | 0.31 | 0.165 | 0.125 | 0.09 |
| | Reinvent | 123 | 4.004 | 254.540 | 0.394 | 0.999 | 1 | 0.0 | 0.728 | 0.000 | 0 | 0 | 0.005 | 0.796 | 0.587 | 0.583 | 0.572 |
| | Frag GFlowNet | 12 | 19.352 | 279.896 | 0.391 | 1 | 1 | 0.0 | 0.723 | 0.000 | 0.196 | 0.003 | 0.243 | 0.565 | 0.531 | 0.517 | 0.472 |
| | Graph MCTS | 0 | 48.600 | 120.600 | 0.309 | 1 | 1 | 0.0 | 0.696 | 0.000 | 0.469 | 0.023 | 0.144 | 0.625 | 0.458 | 0.409 | 0.36 |
| | AGFN Tasktrained | 0 | 0.384 | 0.128 | 0.066 | 1 | 1 | 0.0 | 0.167 | 0.055 | 1.500 | 0.020 | 1.888 | 0.000 | 0.161 | 0.159 | 0.128 |
| | AGFN Finetuned | 0 | 69.055 | 50.508 | 0.074 | 1 | 1 | 0.0 | 0.464 | 0.164 | 1.171 | 1.170 | 3.006 | 0.000 | 0.298 | 0.161 | 0.083 |
| | AGFN FT w/ data | 566 | **318.000** | 823.000 | 0.873 | 1 | 1 | 30.0 | **1.000** | 0.133 | 0.271 | 0.113 | 0.379 | 0.000 | 1.000 | 1.000 | 1.000 |
| | AGFN FT w/ RTB | 688 | 252.469 | **986.923** | 0.853 | 1 | 1 | 28.6 | **1.000** | 0.096 | 0.244 | 0.121 | 0.275 | 0.000 | 1.000 | 1.000 | 0.998 |
| | AGFN FT w/ RTB + data | 567 | 299.000 | 793.000 | **0.879** | 1 | 1 | 42.8 | **1.000** | 0.117 | 0.313 | 0.106 | 0.297 | 0.000 | 1.000 | 1.000 | **1.000** |
| High TPSA | MolDQN | 0 | 18.615 | 18.462 | 0.046 | 1 | 1 | 0.0 | 0.213 | 0.001 | 0.763 | 0.642 | 1.645 | 0.299 | 0.08 | 0.071 | 0.051 |
| | Reinvent | 17 | 5.676 | 293.088 | 0.363 | 0.999 | 1 | 0.0 | 0.684 | 0.000 | 0.001 | 0 | 0.16 | 0.526 | 0.529 | 0.527 | 0.516 |
| | Frag GFlowNet | 0 | 25.705 | 165.625 | 0.229 | 1 | 1 | 0.1 | 0.616 | 0.001 | 0.271 | 0.141 | 0.334 | 0.308 | 0.416 | 0.362 | 0.265 |
| | Graph MCTS | 0 | 36.080 | 60.024 | 0.142 | 1 | 1 | 0.0 | 0.533 | 0.001 | 0.665 | 0.192 | 0.954 | 0.415 | 0.285 | 0.257 | 0.164 |
| | AGFN Tasktrained | 0 | 0.184 | 0.102 | 0.002 | 1 | 1 | 0.0 | 0.076 | 2.158 | 1.370 | 1.850 | 2.594 | 0.000 | 0.070 | 0.011 | 0.002 |
| | AGFN Finetuned | 0 | 19.860 | 12.736 | 0.018 | 1 | 1 | 0.0 | 0.259 | 0.543 | 1.237 | 2.091 | 4.212 | 0.000 | 0.133 | 0.073 | 0.020 |
| | AGFN FT w/ data | 393 | **296.881** | 832.667 | **0.871** | 1 | 1 | 30.1 | **1.000** | 0.380 | 0.259 | 0.162 | 0.325 | 0.000 | 1.000 | 1.000 | **1.000** |
| | AGFN FT w/ RTB | 706 | 231.000 | 985.00 | 0.848 | 1 | 1 | 42.7 | **1.000** | 0.299 | 0.173 | 0.071 | 0.279 | 0.000 | 1.000 | 1.000 | **1.000** |
| | AGFN FT w/ RTB + data | 697 | 203.000 | **989.00** | 0.845 | 1 | 1 | 41.9 | **1.000** | 0.283 | 0.166 | 0.075 | 0.241 | 0.000 | 1.000 | 1.000 | **1.000** |

*Table 15.* Comparing the effectiveness of finetuned A-GFN over A-GFN trained from scratch and other baselines for property-constrained optimization objective for Lipophilicity task.

### F.4.5. TASK: ABSORPTION - SOLUBILITY

| | | Chem. Space Coverage | | Diversity | | | | | MOO metrics | | | | | | Reward | | |
| | | | | | | | | | | L1-dist ↓ | | | | | | | |
| | Method | #modes | RW#C | RW#S | RWTD | Nov | Uniq | Success % | HV | TPSA | #Rings | SAS | QED | Task | Top-1 | Top-10 | Top-100 |
|---|---|---|---|---|---|---|---|---|---|---|---|---|---|---|---|---|---|
| Preserved | MolDQN | 28 | 5.075 | 45.676 | 0.390 | 1 | 1 | 0.5 | 0.332 | 0.000 | 0.106 | 0.026 | 0.153 | 0.396 | 0.643 | 0.619 | 0.564 |
| | Reinvent | 147 | 8.809 | 235.435 | 0.593 | 1 | 1 | 1.6 | 0.438 | 0.000 | 0.000 | 0.000 | 0.146 | 0.436 | 0.825 | 0.816 | **0.801** |
| | Frag GFlowNet | 71 | 21.956 | 319.588 | 0.509 | 1 | 1 | 11.3 | 0.426 | 0.000 | 0.120 | 0.014 | 0.208 | 0.232 | 0.711 | 0.688 | 0.610 |
| | Graph MCTS | 14 | 73.586 | 155.140 | 0.401 | 1 | 1 | 0.2 | 0.388 | 0.000 | 0.485 | 0.052 | 0.153 | 0.265 | 0.640 | 0.553 | 0.469 |
| | AGFN Tasktrained | 0 | 1.000 | 0.920 | 0.009 | 1 | 1 | 0.0 | 0.008 | 1.265 | 1.200 | 2.100 | 4.000 | 1.300 | 0.150 | 0.050 | 0.010 |
| | AGFN Finetuned | 0 | 25.395 | 21.043 | 0.023 | 1 | 1 | 0.0 | 0.352 | 0.076 | 1.072 | 1.866 | 3.693 | 1.181 | 0.251 | 0.098 | 0.026 |
| | AGFN FT w/ data | 234 | **270.205** | **570.906** | **0.623** | 1 | 1 | 0.0 | **0.978** | 0.045 | 0.439 | 0.162 | 0.397 | 1.375 | 0.956 | 0.874 | 0.709 |
| | AGFN FT w/ RTB | 110 | 20.536 | 552.660 | 0.464 | 1 | 1 | 0.0 | 0.829 | 0.029 | 0.305 | 0.026 | 0.454 | 0.427 | 0.707 | 0.669 | 0.604 |
| | AGFN FT w/ RTB + data | 0 | 116.424 | 242.946 | 0.261 | 0 | 1 | 1.0 | 0.602 | 0.040 | 0.513 | 0.163 | 0.404 | 1.194 | 0.493 | 0.430 | 0.297 |
| Low TPSA | MolDQN | 0 | 6.497 | 39.603 | 0.229 | 1 | 1 | 0.0 | 0.309 | 0.000 | 0.565 | 0.193 | 0.158 | 0.215 | 0.473 | 0.410 | 0.309 |
| | Reinvent | 121 | 6.490 | 456.706 | **0.567** | 1 | 1 | 0.7 | 0.477 | 0.000 | 0.000 | 0.000 | 0.005 | 0.365 | 0.832 | 0.827 | **0.811** |
| | Frag GFlowNet | 73 | 47.068 | 422.422 | 0.518 | 1 | 1 | 4.3 | 0.455 | 0.001 | 0.335 | 0.059 | 0.144 | 0.302 | 0.785 | 0.706 | 0.596 |
| | Graph MCTS | 43 | 72.846 | 171.855 | 0.441 | 1 | 1 | 0.0 | 0.458 | 0.000 | 0.502 | 0.022 | 0.159 | 0.239 | 0.638 | 0.586 | 0.513 |
| | AGFN Tasktrained | 0 | 2.160 | 1.100 | 0.017 | 1 | 1 | 0.0 | 0.006 | 1.133 | 1.300 | 1.700 | 3.500 | 1.200 | 0.200 | 0.080 | 0.020 |
| | AGFN Finetuned | 0 | 48.913 | 36.000 | 0.050 | 1 | 1 | 0.0 | 0.473 | 0.206 | 1.199 | 1.302 | 3.122 | 1.077 | 0.355 | 0.143 | 0.056 |
| | AGFN FT w/ data | 139 | **188.292** | 462.885 | 0.527 | 1 | 1 | 0.0 | **0.964** | 0.105 | 0.399 | 0.144 | 0.413 | 1.414 | 0.863 | 0.735 | 0.604 |
| | AGFN FT w/ RTB | 118 | 27.219 | **609.579** | 0.498 | 1 | 1 | 0.0 | 0.851 | 0.093 | 0.298 | 0.059 | 0.258 | 0.416 | 0.735 | 0.700 | 0.633 |
| | AGFN FT w/ RTB + data | 0 | 78.861 | 196.746 | 0.236 | 1 | 1 | 0.0 | 0.693 | 0.122 | 0.501 | 0.092 | 0.365 | 1.208 | 0.457 | 0.387 | 0.271 |
| High TPSA | MolDQN | 0 | 19.936 | 20.527 | 0.036 | 1 | 1 | 0.0 | 0.111 | 0.001 | 0.724 | 0.832 | 1.794 | 0.352 | 0.127 | 0.087 | 0.039 |
| | Reinvent | 50 | 1.986 | 83.399 | 0.435 | 1 | 1 | 0.1 | 0.362 | 0.000 | 0.000 | 0.000 | 0.084 | 0.452 | 0.686 | 0.678 | 0.662 |
| | Frag GFlowNet | 1 | 34.934 | 244.177 | 0.312 | 1 | 1 | 7.2 | 0.329 | 0.001 | 0.224 | 0.132 | 0.279 | 0.163 | 0.502 | 0.464 | 0.364 |
| | Graph MCTS | 0 | 46.759 | 88.056 | 0.189 | 1 | 1 | 0.0 | 0.310 | 0.001 | 0.596 | 0.196 | 0.941 | 0.251 | 0.404 | 0.343 | 0.219 |
| | AGFN Tasktrained | 0 | 1.050 | 1.320 | 0.013 | 1 | 1 | 0.0 | 0.006 | 0.997 | 1.100 | 1.900 | 3.700 | 1.100 | 0.250 | 0.090 | 0.015 |
| | AGFN Finetuned | 0 | 25.572 | 23.104 | 0.023 | 1 | 1 | 0.0 | 0.463 | 0.371 | 1.006 | 1.970 | 3.878 | 1.059 | 0.385 | 0.110 | 0.026 |
| | AGFN FT w/ data | 253 | **151.609** | **439.512** | **0.659** | 1 | 1 | 0.0 | **0.973** | 0.286 | 0.396 | 0.115 | 0.437 | 1.212 | 0.947 | 0.875 | **0.766** |
| | AGFN FT w/ RTB | 0 | 48.158 | 386.458 | 0.327 | 1 | 1 | 0.0 | 0.566 | 0.166 | 0.172 | 0.044 | 0.439 | 1.194 | 0.500 | 0.477 | 0.398 |
| | AGFN FT w/ RTB + data | 0 | 75.816 | 214.488 | 0.277 | 1 | 1 | 0.0 | 0.493 | 0.275 | 0.453 | 0.090 | 0.332 | 1.096 | 0.453 | 0.420 | 0.324 |

*Table 16.* Comparing the effectiveness of finetuned A-GFN over A-GFN trained from scratch and other baselines for property-constrained optimization objective for Aqueous Solubility task.

F.4.6. TASK: DISTRIBUTION - PLASMA PROTEIN BINDING RATE

| | Method | Chem. Space Coverage | | | Diversity | | | | MOO metrics | | | | | | Reward | | |
|---|---|---|---|---|---|---|---|---|---|---|---|---|---|---|---|---|---|
| | | | | | | | | | | | L1-dist ↓ | | | | | | |
| | | #modes | RW#C | RW#S | RWTD | Nov | Uniq | Success % | HV | TPSA | #Rings | SAS | QED | Task | Top-1 | Top-10 | Top-100 |
| Preserved | MolDQN | 0 | 0.002 | 0.002 | 0.000 | 1 | 1 | 0.0 | 0.000 | 0.000 | 0.888 | 0.541 | 1.081 | 0.476 | $< 10^{-4}$ | $< 10^{-4}$ | $< 10^{-4}$ |
| | Reinvent | 0 | 0.013 | 0.062 | 0.000 | 1 | 1 | 0.0 | 0.000 | 0.000 | 0.203 | 0.006 | 0.221 | 0.896 | $< 10^{-4}$ | $< 10^{-4}$ | $< 10^{-4}$ |
| | Frag GFlowNet | 0 | 0.005 | 0.040 | 0.000 | 1 | 1.000 | 0 | 0.000 | 0.0 | 0.178 | 0.051 | 0.204 | 0.792 | $< 10^{-4}$ | $< 10^{-4}$ | $< 10^{-4}$ |
| | Graph MCTS | 0 | 0.007 | 0.017 | 0.000 | 1 | 1.000 | 0 | 0.000 | 0.0 | 0.448 | 0.068 | 0.214 | 0.841 | $< 10^{-4}$ | $< 10^{-4}$ | $< 10^{-4}$ |
| | AGFN Tasktrained | 0 | 0.000 | 0.000 | 0.000 | 1 | 1 | 0.0 | 0.000 | 0.438 | 1.155 | 1.635 | 2.319 | 0.648 | 0.000 | 0.000 | 0.000 |
| | AGFN Finetuned | 0 | 0.001 | 0.001 | 0.000 | 1 | 1 | 0.0 | 0.000 | 0.117 | 1.037 | 2.205 | 4.007 | 0.753 | 0.000 | 0.000 | 0.000 |
| | AGFN FT w/ data | 0 | 0.023 | 0.061 | 0.000 | 1 | 1 | 0.0 | 0.000 | 0.059 | 0.305 | 0.124 | 0.324 | 0.923 | 0.000 | 0.000 | 0.000 |
| | AGFN FT w/ RTB | 136 | 29.210 | **602.615** | **0.488** | 1 | 1 | 0.0 | **0.788** | 0.019 | 0.143 | 0.049 | 0.250 | 0.454 | 0.752 | 0.703 | **0.635** |
| | AGFN FT w/ RTB + data | 0 | **132.683** | 218.830 | 0.244 | 1 | 1 | 0.0 | 0.593 | 0.055 | 0.569 | 0.200 | 0.473 | 0.797 | 0.417 | 0.379 | 0.277 |
| Low TPSA | MolDQN | 0 | 0.001 | 0.001 | 0.000 | 1 | 1 | 0 | 0.000 | 0.000529 | 1.075 | 0.4328 | 1.074 | 0.3948 | $< 10^{-4}$ | $< 10^{-4}$ | $< 10^{-4}$ |
| | Reinvent | 0 | 0.006 | 0.019 | 0.000 | 0.996 | 1 | 0 | 0.000 | 0.000612 | 0.3195 | 0.0478 | 0.301 | 0.8828 | $< 10^{-4}$ | $< 10^{-4}$ | $< 10^{-4}$ |
| | Frag GFlowNet | 0 | 0.002 | 0.011 | 0.000 | 1 | 1 | 0 | 0.000 | 0.001204 | 0.383 | 0.1205 | 0.243 | 0.7692 | $< 10^{-4}$ | $< 10^{-4}$ | $< 10^{-4}$ |
| | Graph MCTS | 0 | 0.003 | 0.007 | 0.000 | 1 | 1 | 0 | 0.000 | 0.000109 | 0.4505 | 0.0435 | 0.182 | 0.8686 | $< 10^{-4}$ | $< 10^{-4}$ | $< 10^{-4}$ |
| | AGFN Tasktrained | 0 | 0.000 | 0.000 | 0.000 | 0 | 0 | 0.0 | 0.000 | 0.000 | 1.500 | 0.036 | 2.383 | 0.399 | 0.000 | 0.000 | 0.000 |
| | AGFN Finetuned | 0 | 0.000 | 0.000 | 0.000 | 1 | 1 | 0.0 | 0.000 | 0.237 | 1.173 | 1.869 | 3.739 | 0.783 | 0.000 | 0.000 | 0.000 |
| | AGFN FT w/ data | 0 | 0.007 | 0.020 | 0.000 | 1 | 1 | 0.0 | 0.000 | 0.161 | 0.305 | 0.105 | 0.414 | 0.971 | 0.000 | 0.000 | 0.000 |
| | AGFN FT w/ RTB | 154 | 26.732 | **611.576** | **0.505** | 1 | 1 | 0.0 | **0.818** | 0.081 | 0.166 | 0.062 | 0.163 | 0.416 | 0.742 | 0.708 | **0.652** |
| | AGFN FT w/ RTB + data | 0 | **136.697** | 234.668 | 0.257 | 1 | 1 | 0.0 | 0.671 | 0.188 | 0.523 | 0.179 | 0.394 | 0.837 | 0.494 | 0.409 | 0.289 |
| High TPSA | MolDQN | 0 | 0.000 | 0.000 | 0.000 | 1 | 1 | 0 | 0.000 | 0.000535 | 0.796 | 0.9074 | 1.844 | 0.376 | $< 10^{-4}$ | $< 10^{-4}$ | $< 10^{-4}$ |
| | Reinvent | 0 | 0.000 | 0.000 | 0.000 | 0.998 | 1 | 0.1 | 0.000 | 0.158375 | 0.384 | 0.1572 | 0.59 | 0.6318 | $< 10^{-4}$ | $< 10^{-4}$ | $< 10^{-4}$ |
| | Frag GFlowNet | 0 | 0.000 | 0.000 | 0.000 | 1 | 1 | 0.7 | 0.000 | 0.001132 | 0.301 | 0.2374 | 0.579 | 0.4716 | $< 10^{-4}$ | $< 10^{-4}$ | $< 10^{-4}$ |
| | Graph MCTS | 0 | 0.000 | 0.000 | 0.000 | 1 | 1 | 0 | 0.000 | 0.000939 | 0.5615 | 0.2006 | 0.869 | 0.4183 | $< 10^{-4}$ | $< 10^{-4}$ | $< 10^{-4}$ |
| | AGFN Tasktrained | 0 | 0.000 | 0.000 | 0.000 | 1 | 1 | 0.0 | 0.000 | 1.451 | 1.270 | 2.019 | 2.534 | 0.304 | 0.000 | 0.000 | 0.000 |
| | AGFN Finetuned | 0 | 0.000 | 0.000 | 0.000 | 1 | 1 | 0.0 | 0.000 | 0.456 | 1.129 | 2.134 | 4.211 | 0.364 | 0.000 | 0.000 | 0.000 |
| | AGFN FT w/ data | 0 | 0.000 | 0.000 | 0.000 | 1 | 1 | 0.1 | 0.000 | 0.524 | 0.281 | 0.162 | 0.377 | 0.715 | 0.000 | 0.000 | 0.000 |
| | AGFN FT w/ RTB | 403 | 64.158 | **818.448** | **0.691** | 1 | 1 | 13.1 | **0.989** | 0.173 | 0.147 | 0.045 | 0.331 | 0.302 | 0.994 | 0.966 | **0.867** |
| | AGFN FT w/ RTB + data | 12 | **200.992** | 325.248 | 0.310 | 0 | 1 | 1.0 | 0.893 | 0.395 | 0.456 | 0.467 | 0.637 | 0.604 | 0.772 | 0.648 | 0.352 |

*Table 17.* Comparing the effectiveness of finetuned A-GFN over A-GFN trained from scratch and other baselines for property-constrained optimization objective for Plasma Protein Binding Rate task.

### F.4.7. Task: Excretion - Hepatocyte Clearance

| | Method | Chem. Space Coverage | | Diversity | | Nov | Uniq | Success % | MOO metrics | | | | | | Reward | | |
|---|---|---|---|---|---|---|---|---|---|---|---|---|---|---|---|---|---|
| | | | | | | | | | HV | L1-dist ↓ | | | | | Top-1 | Top-10 | Top-100 |
| | | #modes | RW#C | RW#S | RWTD | | | | | TPSA | #Rings | SAS | QED | Task | | | |
| Preserved | MolDQN | 0 | 0.806 | 0.697 | 0.002 | 1 | 1 | 0 | 0.022 | $<10^{-4}$ | 0.952 | 0.514 | 1.193 | 2.798 | 0.006 | 0.004 | 0.002 |
| | Reinvent | 0 | 1.895 | 19.732 | 0.020 | 0.999 | 1 | 0 | 0.075 | $<10^{-4}$ | 0.154 | 0.000 | 0.201 | 5.054 | 0.031 | 0.028 | 0.024 |
| | Frag GFlowNet | 0 | 1.210 | 15.245 | 0.016 | 1 | 1 | 0 | 0.060 | $<10^{-4}$ | 0.110 | 0.003 | 0.223 | 2.917 | 0.025 | 0.023 | 0.019 |
| | Graph MCTS | 0 | 2.291 | 4.961 | 0.013 | 1 | 1 | 0 | 0.059 | $<10^{-4}$ | 0.479 | 0.043 | 0.165 | 4.798 | 0.020 | 0.019 | 0.016 |
| | AGFN Tasktrained | 0 | 0.006 | 0.004 | 0.000 | 1 | 1 | 0.0 | 0.000 | 0.865 | 1.165 | 1.980 | 2.994 | 2.405 | 0.001 | 0.000 | 0.000 |
| | AGFN Finetuned | 0 | 0.900 | 0.828 | 0.001 | 1 | 1 | 0.0 | 0.015 | 0.092 | 1.014 | 2.177 | 4.050 | 3.357 | 0.009 | 0.004 | 0.001 |
| | AGFN FT w/ data | 0 | 9.455 | 23.994 | 0.027 | 1 | 1 | 0.3 | 0.073 | 0.071 | 0.280 | 0.141 | 0.326 | 2.335 | 0.075 | 0.050 | 0.031 |
| | AGFN FT w/ RTB | 26 | 9.425 | **608.275** | **0.468** | 1 | 1 | 1.3 | **0.993** | 0.063 | 0.092 | 0.077 | 2.026 | 0.423 | 0.942 | 0.894 | **0.725** |
| | AGFN FT w/ RTB + data | 9 | **79.728** | 152.944 | 0.154 | 1 | 1 | 0.2 | 0.935 | 0.064 | 0.380 | 0.244 | 0.430 | 2.202 | 0.772 | 0.681 | 0.176 |
| Low TPSA | MolDQN | 0 | 0.706 | 0.575 | 0.002 | 1 | 1 | 0 | 0.020 | $<10^{-4}$ | 0.952 | 0.514 | 1.193 | 2.798 | 0.006 | 0.004 | 0.002 |
| | Reinvent | 0 | 1.701 | 18.881 | 0.020 | 0.995 | 1 | 0 | 0.077 | $<10^{-4}$ | 0.154 | 0.000 | 0.201 | 5.054 | 0.031 | 0.028 | 0.024 |
| | Frag GFlowNet | 0 | 1.862 | 14.515 | 0.017 | 1 | 1 | 0 | 0.060 | $<10^{-4}$ | 0.110 | 0.003 | 0.223 | 2.917 | 0.025 | 0.023 | 0.019 |
| | Graph MCTS | 0 | 2.275 | 5.009 | 0.014 | 1 | 1 | 0 | 0.071 | $<10^{-4}$ | 0.479 | 0.043 | 0.165 | 4.798 | 0.020 | 0.019 | 0.016 |
| | AGFN Tasktrained | 0 | 0.003 | 0.003 | 0.000 | 0 | 0 | 0.0 | 0.003 | 0.000 | 1.500 | 0.036 | 2.383 | 0.081 | 0.003 | 0.003 | 0.003 |
| | AGFN Finetuned | 0 | 0.797 | 0.664 | 0.001 | 1 | 1 | 0.0 | 0.009 | 0.223 | 1.095 | 1.934 | 3.760 | 0.308 | 0.003 | 0.002 | 0.001 |
| | AGFN FT w/ data | 0 | 11.837 | 31.318 | 0.036 | 1 | 1 | 2.4 | 0.103 | 0.181 | 0.274 | 0.126 | 0.503 | 0.417 | 0.104 | 0.062 | 0.041 |
| | AGFN FT w/ RTB | 192 | 29.845 | **779.486** | **0.633** | 1 | 1 | 6.5 | **0.992** | 0.058 | 0.228 | 0.027 | 0.117 | 0.266 | 0.990 | 0.968 | **0.878** |
| | AGFN FT w/ RTB + data | 124 | **254.100** | 594.00 | 0.574 | 1 | 1 | 1.2 | 0.980 | 0.175 | 0.298 | 0.284 | 1.193 | 0.292 | 0.949 | 0.854 | 0.660 |
| High TPSA | MolDQN | 0 | 0.081 | 0.072 | 0.001 | 1 | 1 | 0.0 | 0.011 | 0.000 | 0.730 | 0.565 | 1.384 | 1.523 | 0.003 | 0.002 | 0.001 |
| | Reinvent | 0 | 0.565 | 1.165 | 0.010 | 1 | 1 | 0.0 | 0.055 | 0.000 | 0.190 | 0.010 | 0.267 | 2.259 | 0.020 | 0.017 | 0.012 |
| | Frag GFlowNet | 0 | 0.319 | 1.027 | 0.010 | 1 | 1 | 0.0 | 0.045 | 0.000 | 0.155 | 0.022 | 0.078 | 1.436 | 0.018 | 0.016 | 0.012 |
| | Graph MCTS | 0 | 0.228 | 0.396 | 0.006 | 1 | 1 | 0.0 | 0.046 | 0.000 | 0.440 | 0.067 | 0.265 | 2.601 | 0.015 | 0.011 | 0.007 |
| | AGFN Tasktrained | 0 | 0.004 | 0.003 | 0.000 | 1 | 1 | 0.0 | 0.002 | 2.035 | 1.220 | 2.033 | 2.443 | 1.238 | 0.002 | 0.000 | 0.000 |
| | AGFN Finetuned | 0 | 0.031 | 0.031 | 0.002 | 1 | 1 | 0.0 | 0.006 | 0.337 | 1.500 | 0.255 | 3.933 | 2.463 | 0.005 | 0.004 | 0.004 |
| | AGFN FT w/ data | 0 | 3.792 | 11.439 | 0.025 | 1 | 1 | 3.9 | 0.048 | 0.580 | 0.180 | 0.090 | 0.303 | 0.886 | 0.045 | 0.042 | 0.032 |
| | AGFN FT w/ RTB | 104 | 13.813 | **710.938** | **0.581** | 1 | 1 | 5.9 | **0.991** | 0.205 | 0.121 | 0.084 | 0.723 | 0.363 | 0.983 | 0.938 | **0.813** |
| | AGFN FT w/ RTB + data | 37 | **237.442** | 391.802 | 0.400 | 1 | 1 | 1.0 | 0.981 | 0.543 | 0.426 | 0.431 | 0.656 | 0.817 | 0.964 | 0.836 | 0.454 |

*Table 18.* Comparing the effectiveness of finetuned A-GFN over A-GFN trained from scratch and other baselines for property-constrained optimization objective for Hepatocyte Clearance task.

## F.4.8. TASK: EXCRETION - MICROSOMAL CLEARANCE

| | Method | Chem. Space Coverage | | Diversity | | | | | MOO metrics | | | | | | Reward | | |
| | | | | | | | | | | L1-dist ↓ | | | | | | | |
| | | #modes | RW#C | RW#S | RWTD | Nov | Uniq | Success % | HV | TPSA | #Rings | SAS | QED | Task | Top-1 | Top-10 | Top-100 |
|---|---|---|---|---|---|---|---|---|---|---|---|---|---|---|---|---|---|
| Preserved | MolDQN | 0 | 0.116 | 0.102 | 0.000 | 1 | 1 | 0 | 0.003 | 0.000 | 0.928 | 0.538 | 1.202 | 1.022 | 0.001 | 0.001 | 0.000 |
| | Reinvent | 0 | 0.468 | 2.285 | 0.002 | 0.998 | 1 | 0 | 0.007 | 0.000 | 0.240 | 0.008 | 0.222 | 2.094 | 0.003 | 0.003 | 0.003 |
| | Frag GFlowNet | 0 | 0.176 | 1.727 | 0.002 | 1 | 1 | 0.2 | 0.007 | 0.000 | 0.194 | 0.027 | 0.172 | 1.495 | 0.003 | 0.003 | 0.002 |
| | Graph MCTS | 0 | 0.312 | 0.688 | 0.002 | 1 | 1 | 0 | 0.007 | 0.000 | 0.478 | 0.056 | 0.153 | 1.910 | 0.003 | 0.002 | 0.002 |
| | AGFN Tasktrained | 0 | 0.004 | 0.002 | 0.000 | 1 | 1 | 0.0 | 0.002 | 0.532 | 1.275 | 1.735 | 1.885 | 1.330 | 0.001 | 0.000 | 0.000 |
| | AGFN Finetuned | 0 | 0.399 | 0.348 | 0.000 | 1 | 1 | 0.0 | 0.005 | 0.119 | 1.045 | 2.155 | 4.189 | 3.976 | 0.004 | 0.002 | 0.000 |
| | AGFN FT w/ data | 0 | 2.723 | 6.601 | 0.006 | 1 | 1 | 6.0 | 0.014 | 0.074 | 0.258 | 0.172 | 0.440 | 1.463 | 0.012 | 0.010 | 0.007 |
| | AGFN FT w/ RTB | 0 | 2.758 | **10.704** | **0.009** | 1 | 1 | 0.3 | **0.019** | 0.021 | 0.190 | 0.076 | 0.252 | 2.551 | 0.017 | 0.014 | **0.011** |
| | AGFN FT w/ RTB + data | 0 | **2.993** | 6.512 | 0.006 | 1 | 1 | 5.0 | 0.013 | 0.052 | 0.322 | 0.174 | 0.334 | 1.329 | 0.013 | 0.010 | 0.007 |
| Low TPSA | MolDQN | 0 | 0.102 | 0.095 | 0.000 | 1 | 1 | 0 | 0.003 | 0.000 | 1.063 | 0.430 | 1.076 | 0.849 | 0.001 | 0.001 | 0.000 |
| | Reinvent | 0 | 0.504 | 2.309 | 0.002 | 1 | 1 | 0 | 0.007 | 0.000 | 0.285 | 0.006 | 0.305 | 2.146 | 0.003 | 0.003 | 0.003 |
| | Frag GFlowNet | 0 | 0.250 | 1.409 | 0.002 | 1 | 1 | 0 | 0.007 | 0.001 | 0.354 | 0.100 | 0.171 | 1.743 | 0.003 | 0.003 | 0.002 |
| | Graph MCTS | 0 | 0.290 | 0.748 | 0.002 | 1 | 1 | 0 | 0.007 | 0.000 | 0.459 | 0.021 | 0.150 | 2.204 | 0.003 | 0.002 | 0.002 |
| | AGFN Tasktrained | 0 | 0.006 | 0.003 | 0.000 | 1 | 1 | 0.0 | 0.002 | 0.820 | 1.470 | 0.451 | 0.445 | 0.728 | 0.001 | 0.001 | 0.000 |
| | AGFN Finetuned | 0 | 0.299 | 0.269 | 0.000 | 1 | 1 | 0.0 | 0.005 | 0.259 | 0.965 | 2.210 | 4.191 | 3.848 | 0.003 | 0.001 | 0.000 |
| | AGFN FT w/ data | 0 | 1.659 | 4.673 | 0.005 | 1 | 1 | 5.6 | 0.014 | 0.160 | 0.268 | 0.132 | 0.670 | 1.017 | 0.009 | 0.008 | 0.006 |
| | AGFN FT w/ RTB | 0 | **2.496** | **10.254** | **0.009** | 1 | 1 | 0.1 | **0.018** | 0.155 | 0.233 | 0.118 | 0.253 | 2.822 | 0.017 | 0.015 | **0.010** |
| | AGFN FT w/ RTB + data | 0 | 2.114 | 5.124 | 0.006 | 1 | 1 | 6.4 | 0.013 | 0.169 | 0.334 | 0.126 | 0.359 | 0.996 | 0.014 | 0.010 | 0.007 |
| High TPSA | MolDQN | 0 | 0.107 | 0.108 | 0.000 | 1 | 1 | 0 | 0.002 | 0.001 | 0.761 | 0.892 | 1.849 | 1.393 | 0.001 | 0.000 | 0.000 |
| | Reinvent | 0 | 0.530 | 1.386 | 0.001 | 0.998 | 1 | 0 | 0.007 | 0.088 | 0.397 | 0.167 | 0.623 | 2.310 | 0.003 | 0.002 | 0.002 |
| | Frag GFlowNet | 0 | 0.174 | 1.213 | 0.001 | 1 | 1 | 0.4 | 0.005 | 0.001 | 0.192 | 0.135 | 0.320 | 1.272 | 0.002 | 0.002 | 0.002 |
| | Graph MCTS | 0 | 0.197 | 0.353 | 0.001 | 1 | 1 | 0 | 0.005 | 0.001 | 0.652 | 0.171 | 0.862 | 1.632 | 0.002 | 0.002 | 0.001 |
| | AGFN Tasktrained | 0 | 0.001 | 0.000 | 0.000 | 1 | 1 | 0.0 | 0.000 | 2.904 | 1.355 | 1.755 | 2.301 | 1.119 | 0.000 | 0.000 | 0.000 |
| | AGFN Finetuned | 0 | 0.029 | 0.054 | 0.001 | 1 | 1 | 0.0 | 0.003 | 0.366 | 1.473 | 0.564 | 4.097 | 4.453 | 0.001 | 0.001 | 0.001 |
| | AGFN FT w/ data | 0 | 1.159 | 3.931 | 0.005 | 1 | 1 | 4.4 | **0.017** | 0.548 | 0.194 | 0.204 | 0.723 | 1.498 | 0.013 | 0.009 | 0.006 |
| | AGFN FT w/ RTB | 0 | **2.455** | **9.722** | **0.008** | 1 | 1 | 0.3 | **0.017** | 0.217 | 0.153 | 0.111 | 0.258 | 2.586 | 0.015 | 0.013 | **0.010** |
| | AGFN FT w/ RTB + data | 0 | 2.208 | 4.397 | 0.004 | 1 | 1 | 1.1 | 0.011 | 0.720 | 0.482 | 0.377 | 0.478 | 1.412 | 0.010 | 0.007 | 0.005 |

*Table 19.* Comparing the effectiveness of finetuned A-GFN over A-GFN trained from scratch and other baselines for property-constrained optimization objective for Microsomal Clearance task.

F.4.9. TASK: TOXICITY - ACUTE TOXICITY

| | Method | #modes | Chem. Space Coverage RW#C | Diversity RW#S | RWTD | Nov | Uniq | Success % | HV | TPSA | #Rings | SAS | QED | Task | Top-1 | Top-10 | Top-100 |
|---|---|---|---|---|---|---|---|---|---|---|---|---|---|---|---|---|---|
| Preserved | MolDQN | 0 | 2.160 | 1.938 | 0.006 | 1 | 1 | 0.0 | 0.043 | 0.000 | 0.950 | 0.507 | 1.224 | 0.077 | 0.013 | 0.010 | 0.006 |
| | Reinvent | 0 | 0.531 | 23.954 | 0.043 | 1 | 1 | 48.6 | 0.133 | 0.000 | 0.000 | 0.000 | 0.149 | 0.078 | 0.061 | 0.061 | 0.059 |
| | Frag GFlowNet | 0 | 2.904 | 33.440 | 0.038 | 1 | 1 | 43.7 | 0.117 | 0.000 | 0.145 | 0.011 | 0.184 | 0.052 | 0.054 | 0.052 | 0.044 |
| | Graph MCTS | 0 | 5.236 | 11.424 | 0.029 | 1 | 1 | 4.6 | 0.000 | 0.000 | 0.478 | 0.045 | 0.145 | 0.050 | 0.039 | 0.038 | 0.034 |
| | AGFN Tasktrained | 0 | 1.232 | 1.980 | 0.037 | 1 | 1 | 0.0 | 0.000 | 0.000 | 0.000 | 0.010 | 0.199 | 0.040 | 0.039 | 0.035 | 0.044 |
| | AGFN Finetuned | 0 | 43.824 | 39.204 | 0.040 | 1 | 1 | 0.0 | 0.056 | 0.083 | 1.022 | 2.082 | 3.989 | 0.813 | 0.039 | 0.035 | 0.044 |
| | AGFN FT w/ data | 0 | 23.681 | 50.932 | 0.052 | 1 | 1 | 0.0 | 0.124 | 0.041 | 0.409 | 0.216 | 0.492 | 0.929 | 0.089 | 0.081 | 0.060 |
| | AGFN FT w/ RTB | 0 | 17.892 | 271.220 | 0.228 | 1 | 1 | 0.0 | 0.437 | 0.017 | 0.210 | 0.053 | 0.274 | 0.792 | 0.398 | 0.350 | 0.284 |
| | AGFN FT w/ RTB + data | 0 | **143.416** | **288.288** | **0.319** | 1 | 1 | 0.0 | **0.485** | 0.044 | 0.472 | 0.166 | 0.511 | 0.946 | 0.493 | 0.475 | **0.364** |
| Low TPSA | MolDQN | 0 | 1.698 | 1.665 | 0.003 | 1 | 1 | 0.0 | 0.040 | 0.001 | 0.770 | 0.906 | 1.837 | 0.140 | 0.007 | 0.005 | 0.003 |
| | Reinvent | 0 | 2.277 | 50.853 | 0.053 | 1 | 1 | 13.2 | 0.124 | 0.000 | 0.015 | 0.000 | 0.131 | 0.180 | 0.074 | 0.073 | 0.069 |
| | Frag GFlowNet | 0 | 4.180 | 26.448 | 0.033 | 1 | 1 | 3.7 | 0.120 | 0.001 | 0.262 | 0.152 | 0.359 | 0.147 | 0.052 | 0.050 | 0.038 |
| | Graph MCTS | 0 | 4.906 | 8.756 | 0.019 | 1 | 1 | 0.2 | 0.106 | 0.001 | 0.634 | 0.205 | 0.849 | 0.148 | 0.036 | 0.033 | 0.022 |
| | AGFN Tasktrained | 0 | 0.004 | 0.004 | 0.000 | 0 | 0 | 0.0 | 0.004 | 0.000 | 1.500 | 0.036 | 2.383 | 1.102 | 0.004 | 0.004 | 0.004 |
| | AGFN Finetuned | 0 | 3.976 | 3.768 | 0.004 | 1 | 1 | 0.0 | 0.081 | 0.284 | 0.926 | 2.082 | 3.551 | 0.802 | 0.004 | 0.004 | 0.004 |
| | AGFN FT w/ data | 0 | 22.827 | 51.504 | 0.057 | 1 | 1 | 0.0 | 0.168 | 0.135 | 0.374 | 0.175 | 0.517 | 0.928 | 0.159 | 0.101 | 0.064 |
| | AGFN FT w/ RTB | 0 | 12.989 | **392.184** | 0.322 | 1 | 1 | 0.0 | **0.500** | 0.084 | 0.046 | 0.021 | 0.904 | 1.092 | 0.498 | 0.471 | **0.419** |
| | AGFN FT w/ RTB + data | 0 | **106.260** | 249.480 | **0.336** | 1 | 1 | 0.0 | 0.490 | 0.128 | 0.516 | 0.061 | 0.336 | 1.000 | 0.500 | 0.476 | 0.385 |
| High TPSA | MolDQN | 0 | 1.680 | 1.490 | 0.005 | 1 | 1 | 0.0 | 0.031 | 0.000 | 1.067 | 0.430 | 1.113 | 0.104 | 0.010 | 0.008 | 0.005 |
| | Reinvent | 0 | 0.864 | 19.170 | 0.041 | 1 | 1 | 79.8 | 0.162 | 0.000 | 0.000 | 0.000 | 0.159 | 0.051 | 0.056 | 0.056 | 0.054 |
| | Frag GFlowNet | 0 | 3.403 | 30.832 | 0.036 | 1 | 1 | 34.3 | 0.121 | 0.000 | 0.246 | 0.030 | 0.169 | 0.051 | 0.051 | 0.049 | 0.041 |
| | Graph MCTS | 0 | 5.148 | 11.451 | 0.028 | 1 | 1 | 4.1 | 0.117 | 0.000 | 0.492 | 0.031 | 0.155 | 0.052 | 0.039 | 0.037 | 0.033 |
| | AGFN Tasktrained | 0 | 0.000 | 0.000 | 0.000 | 1 | 1 | 0.0 | 0.023 | 1.328 | 1.250 | 1.981 | 2.975 | 0.625 | 0.016 | 0.003 | 0.000 |
| | AGFN Finetuned | 0 | 2.695 | 2.125 | 0.002 | 1 | 1 | 0.0 | 0.036 | 0.417 | 1.146 | 2.131 | 4.202 | 0.651 | 0.027 | 0.013 | 0.003 |
| | AGFN FT w/ data | 0 | 16.212 | 45.394 | 0.050 | 1 | 1 | 0.0 | 0.118 | 0.367 | 0.408 | 0.172 | 0.475 | 0.883 | 0.092 | 0.076 | 0.058 |
| | AGFN FT w/ RTB | 0 | **20.145** | **377.225** | **0.314** | 1 | 1 | 0.0 | **0.520** | 0.146 | 0.153 | 0.062 | 0.361 | 0.687 | 0.479 | 0.461 | **0.395** |
| | AGFN FT w/ RTB + data | 0 | 11.979 | 29.304 | 0.043 | 1 | 1 | 0.0 | 0.113 | 0.426 | 0.491 | 0.142 | 0.397 | 0.907 | 0.065 | 0.062 | 0.050 |

*Table 20.* Comparing the effectiveness of finetuned A-GFN over A-GFN trained from scratch and other baselines for property-constrained optimization objective for Acute Toxicity task.

## F.5. *De novo* Generation

Analogous to contemporary works (Lee et al., 2023b;a), we generate novel molecules for the aforementioned five targets, optimizing for high docking score (DS), QED, and SAS. As in Saturn (Guo & Schwaller, 2024b), benchmarking is conducted using QuickVina2 (Alhossary et al., 2015) for docking, and hit ratio and novel hit ratio are reported §E. Finetuning is performed with DRA property constrained optimization with #rings modified to $[1, 5]$, and $d_p = 1$.

We consider 2 settings for the A-GFN prior - (i) pretrained with solely an offline data, (ii) pretrained with the proposed hybrid online-offline strategy (fig.6). The offline prior is trained on Enamine 6.75M dataset on 4 Nvidia A100-40GB for 18 days, while the hybrid prior is trained on the same setup for 12 days on Zinc250K dataset. With either prior, we observe that an exploitative A-GFN finetuned with high inverse temperature for sampling (high $\beta$) matches or outperforms Saturn and other baselines for the hit ratio metric (Tab.21). However, when structural novelty is enforced via a Tanimoto similarity (T.S.) threshold of 0.4, the offline prior underperforms in novel hit ratio compared to the best Saturn model. This is expected as RTB training objective keeps the finetuned policy anchored to the pretrained policy, the generated molecules retain structural similarities to the training set, particularly when the pretrained model is derived exclusively from offline data. While a T.S. > 0.4 threshold is often used to assess novelty, we argue that it may not always align with practical drug discovery objectives. In many real-world applications, compounds with known scaffolds and well-characterized chemistry are preferable to highly novel structures. Thus, rigid novelty constraints might not always be desirable.

That said, if high structural novelty is essential for a project, finetuning the hybrid prior (offline + online) leads to broader exploration of the chemical space, yielding a significantly higher novel hit ratio while still maintaining strong hit rates, and comprehensively outperforming all baseline methods (table 6 and 22). This again reinforces that incorporating both offline expertise and online experience during pretraining enables the model to generate more diverse yet biologically relevant compounds.

For 5 protein targets, we finetune the pretrained A-GFN model with RTB objective in purely online fashion. Benchmarking results are taken from Saturn (Guo & Schwaller, 2024b). A-GFN is run across 3 seeds. The variability in QuickVina's docking results is accounted by considering the best docking score for each protein-ligand pair across three independent docking runs. The offline prior used is trained on Enamine 6.75M dataset on 4 Nvidia A100-40GB for 18 days, while the hybrid prior is trained on the same setup for 12 days on Zinc250K dataset.

| Method | parp1 | fa7 | 5ht1b | braf | jak2 |
|---|---|---|---|---|---|
| **Datasets** | | | | | |
| ZINC 250k (Sterling & Irwin, 2015) | $3.993 \pm 0.355$ | $1.097 \pm 0.192$ | $24.260 \pm 0.622$ | $1.020 \pm 0.193$ | $6.183 \pm 0.344$ |
| ChEMBL 33 (Gaulton et al., 2012) | $6.077 \pm 0.453$ | $1.830 \pm 0.240$ | $24.163 \pm 0.715$ | $2.073 \pm 0.181$ | $9.013 \pm 0.562$ |
| **Generative Models** | | | | | |
| REINVENT (Olivecrona et al., 2017)) | $4.693 \pm 1.776$ | $1.967 \pm 0.661$ | $26.047 \pm 2.497$ | $2.207 \pm 0.800$ | $5.667 \pm 1.067$ |
| JT-VAE (Jin et al., 2018) | $3.200 \pm 0.348$ | $0.933 \pm 0.152$ | $18.044 \pm 0.747$ | $0.644 \pm 0.157$ | $5.856 \pm 0.204$ |
| GraphAF (Shi et al., 2020a) | $0.822 \pm 0.113$ | $0.011 \pm 0.016$ | $6.978 \pm 0.952$ | $1.422 \pm 0.556$ | $1.233 \pm 0.284$ |
| MORLD (Jeon & Kim, 2020) | $0.047 \pm 0.050$ | $0.007 \pm 0.013$ | $0.893 \pm 0.758$ | $0.047 \pm 0.040$ | $0.227 \pm 0.118$ |
| HierVAE (Jin et al., 2020a) | $1.180 \pm 0.182$ | $0.033 \pm 0.030$ | $0.740 \pm 0.371$ | $0.367 \pm 0.187$ | $0.487 \pm 0.183$ |
| GraphDF (Luo et al., 2021) | $0.044 \pm 0.031$ | $0.000 \pm 0.000$ | $0.000 \pm 0.000$ | $0.016 \pm 0.016$ | $0.011 \pm 0.016$ |
| FREED (Yang et al., 2021) | $4.860 \pm 1.415$ | $1.487 \pm 0.242$ | $14.227 \pm 5.116$ | $2.707 \pm 0.721$ | $6.067 \pm 0.790$ |
| FREED-QS (Yang et al., 2021) | $5.960 \pm 0.902$ | $1.687 \pm 0.177$ | $23.140 \pm 2.422$ | $3.880 \pm 0.623$ | $7.653 \pm 1.373$ |
| LIMO (Eckmann et al., 2022) | $0.456 \pm 0.057$ | $0.443 \pm 0.016$ | $1.200 \pm 0.178$ | $0.278 \pm 0.134$ | $0.711 \pm 0.329$ |
| GDSS (Jo et al., 2022) | $2.367 \pm 0.316$ | $0.467 \pm 0.112$ | $6.267 \pm 0.287$ | $0.300 \pm 0.198$ | $1.367 \pm 0.258$ |
| MOOD (Lee et al., 2023a) | $7.260 \pm 0.764$ | $0.787 \pm 0.122$ | $21.427 \pm 0.502$ | $0.784 \pm 0.311$ | $10.367 \pm 0.616$ |
| Aug. Mem. (Guo & Schwaller, 2024a) | $16.966 \pm 3.224$ | $2.637 \pm 0.860$ | $52.016 \pm 2.302$ | $8.307 \pm 1.714$ | $21.548 \pm 4.938$ |
| GEAM (Lee et al., 2023b) | $45.158 \pm 2.408$ | $20.552 \pm 2.357$ | $47.664 \pm 1.198$ | $30.444 \pm 1.610$ | $46.129 \pm 2.073$ |
| Saturn (Guo & Schwaller, 2024b) | $57.981 \pm 18.537$ | $14.527 \pm 9.961$ | $68.185 \pm 3.400$ | $38.999 \pm 10.114$ | $60.827 \pm 11.502$ |
| **A-GFN Finetuning** | | | | | |
| $\beta = 96$ (offline prior) | $42.900 \pm 2.300$ | $20.167 \pm 3.200$ | $59.200 \pm 2.500$ | $35.334 \pm 3.100$ | $63.434 \pm 2.700$ |
| $\beta = 96$ w/ uniq. filter (offline prior) | $72.507 \pm 3.400$ | $34.181 \pm 2.800$ | $\mathbf{98.175 \pm 3.200}$ | $60.468 \pm 2.900$ | $85.566 \pm 2.500$ |
| $\beta = 96$ (offline + online prior) | $45.800 \pm 2.500$ | $9.200 \pm 4.000$ | $73.400 \pm 2.200$ | $13.500 \pm 3.300$ | $73.400 \pm 2.400$ |
| $\beta = 96$ **w/ uniq. filter (offline + online prior)** | $\mathbf{76.375 \pm 3.200}$ | $\mathbf{36.078 \pm 2.400}$ | $97.910 \pm 3.500$ | $\mathbf{63.579 \pm 2.600}$ | $\mathbf{97.910 \pm 3.400}$ |

*Table 21.* Hit Ratio (%): Finetuning is done with RTB objective in all A-GFN experiments. Unique Filter refers to sampling $n$ unique trajectories ($n = 3000$).

| Method | parp1 | fa7 | 5ht1b | braf | jak2 |
|---|---|---|---|---|---|
| REINVENT (Olivecrona et al., 2017) | $0.480 \pm 0.344$ | $0.213 \pm 0.081$ | $2.453 \pm 0.561$ | $0.127 \pm 0.088$ | $0.613 \pm 0.167$ |
| GCPN (You et al., 2018) | $0.056 \pm 0.016$ | $0.444 \pm 0.333$ | $0.444 \pm 0.150$ | $0.033 \pm 0.027$ | $0.256 \pm 0.087$ |
| JT-VAE (Jin et al., 2018) | $0.856 \pm 0.211$ | $0.289 \pm 0.016$ | $4.656 \pm 1.406$ | $0.144 \pm 0.068$ | $0.815 \pm 0.044$ |
| GraphAF (Shi et al., 2020a) | $0.689 \pm 0.166$ | $0.011 \pm 0.016$ | $3.178 \pm 0.393$ | $0.956 \pm 0.319$ | $0.767 \pm 0.098$ |
| GraphGA (Jensen, 2019) | $4.811 \pm 1.661$ | $0.422 \pm 1.193$ | $7.011 \pm 2.732$ | $3.767 \pm 1.498$ | $5.311 \pm 1.667$ |
| MORLD (Jeon & Kim, 2020) | $0.047 \pm 0.050$ | $0.007 \pm 0.013$ | $0.880 \pm 0.735$ | $0.047 \pm 0.040$ | $0.227 \pm 0.118$ |
| HierVAE (Jin et al., 2020a) | $0.553 \pm 0.214$ | $0.007 \pm 0.013$ | $0.507 \pm 0.278$ | $0.207 \pm 0.220$ | $0.227 \pm 0.127$ |
| RationaleRL (Jin et al., 2020b) | $4.267 \pm 0.450$ | $0.900 \pm 0.096$ | $2.967 \pm 0.307$ | $0.000 \pm 0.000$ | $2.967 \pm 0.196$ |
| GA+D (Nigam et al., 2019) | $0.044 \pm 0.042$ | $0.011 \pm 0.016$ | $1.544 \pm 0.273$ | $0.800 \pm 0.864$ | $0.756 \pm 0.204$ |
| MARS (Xie et al., 2021a) | $1.178 \pm 0.299$ | $0.367 \pm 0.072$ | $6.833 \pm 0.706$ | $0.478 \pm 0.083$ | $2.178 \pm 0.545$ |
| GEGL (Ahn et al., 2020) | $0.789 \pm 0.150$ | $0.256 \pm 0.083$ | $3.167 \pm 0.260$ | $0.244 \pm 0.016$ | $0.933 \pm 0.072$ |
| GraphDF (Luo et al., 2021) | $0.044 \pm 0.031$ | $0.000 \pm 0.000$ | $0.000 \pm 0.000$ | $0.011 \pm 0.016$ | $0.011 \pm 0.016$ |
| FREED (Yang et al., 2021) | $4.627 \pm 0.727$ | $1.332 \pm 0.113$ | $16.767 \pm 0.897$ | $2.940 \pm 0.359$ | $5.800 \pm 0.295$ |
| LIMO (Eckmann et al., 2022) | $0.455 \pm 0.057$ | $0.044 \pm 0.016$ | $1.189 \pm 0.181$ | $0.278 \pm 0.134$ | $0.689 \pm 0.319$ |
| GDSS (Jo et al., 2022) | $1.933 \pm 0.208$ | $0.368 \pm 0.103$ | $4.667 \pm 0.306$ | $0.167 \pm 0.134$ | $1.167 \pm 0.281$ |
| PS-VAE (Kong et al., 2022) | $1.644 \pm 0.389$ | $0.478 \pm 0.140$ | $12.622 \pm 1.437$ | $0.367 \pm 0.047$ | $4.178 \pm 0.933$ |
| MOOD (Lee et al., 2023a) | $7.017 \pm 0.428$ | $0.733 \pm 0.141$ | $18.673 \pm 0.423$ | $5.240 \pm 0.285$ | $9.200 \pm 0.524$ |
| GEAM (Lee et al., 2023b) | $39.159 \pm 2.790$ | $19.540 \pm 2.347$ | $40.123 \pm 1.611$ | $27.467 \pm 1.374$ | $41.765 \pm 3.412$ |
| Saturn (Guo & Schwaller, 2024b) | $3.839 \pm 3.316$ | $0.470 \pm 0.272$ | $5.731 \pm 6.166$ | $3.652 \pm 3.777$ | $6.129 \pm 5.449$ |
| Saturn-Tanimoto (Guo & Schwaller, 2024b) | $50.552 \pm 9.530$ | $20.181 \pm 5.598$ | $54.260 \pm 6.722$ | $19.820 \pm 10.120$ | $47.785 \pm 14.041$ |
| **A-GFN Finetuning** | | | | | |
| $\beta = 96$ (offline prior) | $18.400 \pm 2.300$ | $8.233 \pm 3.200$ | $25.834 \pm 2.500$ | $14.400 \pm 3.100$ | $36.266 \pm 2.700$ |
| $\beta = 96$ w/ uniq. filter (offline prior) | $31.098 \pm 3.400$ | $13.955 \pm 2.800$ | $42.841 \pm 3.200$ | $24.640 \pm 2.900$ | $48.921 \pm 2.500$ |
| $\beta = 96$ (offline + online prior) | $41.734 \pm 2.500$ | $6.934 \pm 4.000$ | $55.267 \pm 2.200$ | $12.134 \pm 3.300$ | $55.267 \pm 2.400$ |
| $\beta = 96$ **w/ uniq. filter (offline + online prior)** | $\mathbf{69.594 \pm 3.400}$ | $\mathbf{27.190 \pm 2.800}$ | $\mathbf{73.721 \pm 3.200}$ | $\mathbf{57.143 \pm 2.900}$ | $\mathbf{73.721 \pm 2.500}$ |

*Table 22.* Novel Hit Ratio (%)

## F.6. Lead Optimization

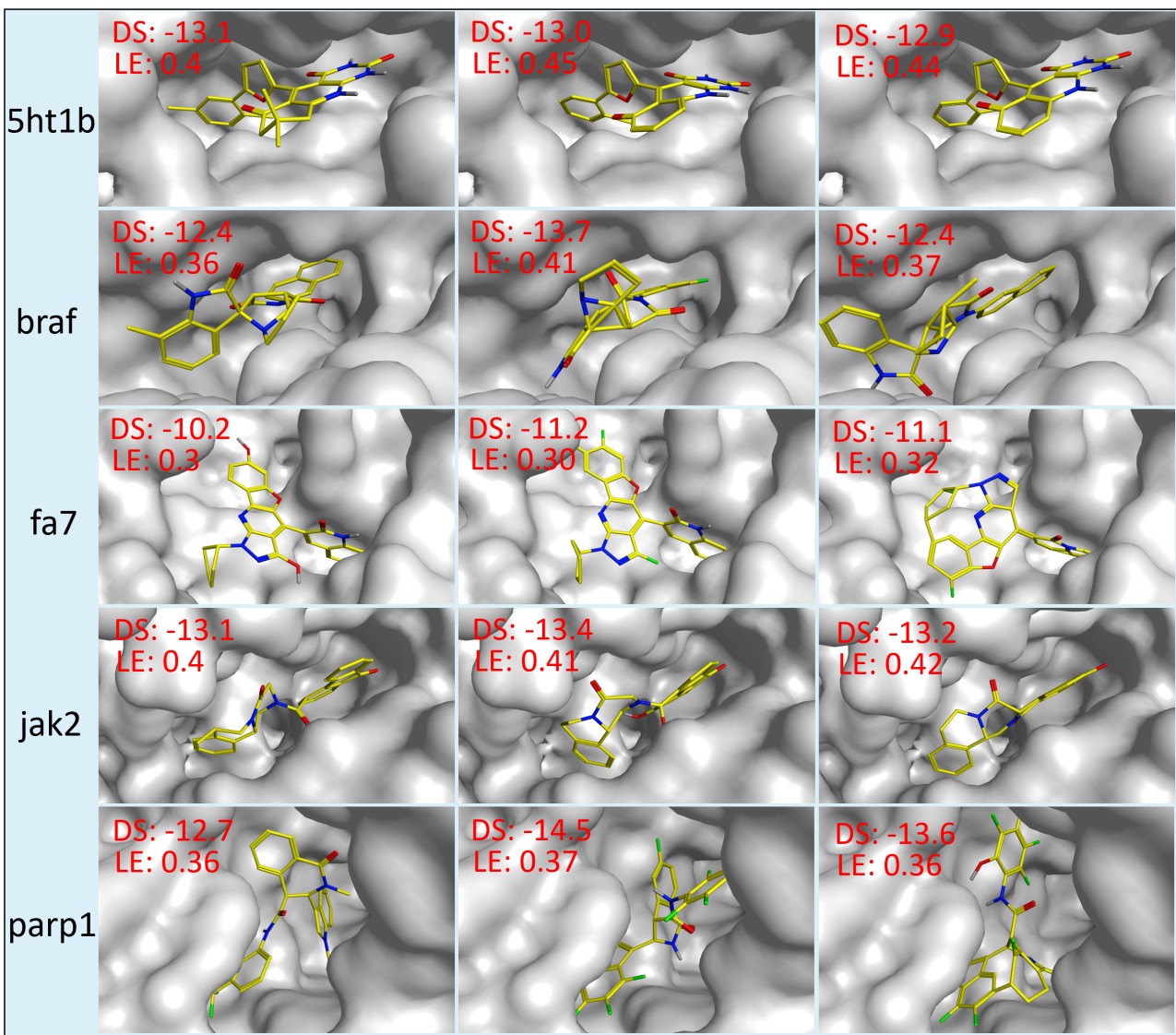

*Figure 11.* Docking comparison of the lead compound and the two best lead-optimized compounds to their respective targets across five targets. First col. shows the chosen lead compound and col.1-2 are optimized for docking score and ligand efficiency.

