# OpenReview forum: "Pretraining Generative Flow Networks with Inexpensive Rewards for Molecular Graph Generation"
_ICML.cc/2025/Conference — ICML 2025 poster_

### Official Review · Reviewer_uZjw · 2025-02-28

**Overall Recommendation:** 3

**Summary:**

The paper introduces Atomic GFlowNets (A-GFN), a novel generative model for molecular graph generation that leverages individual atoms as building blocks to explore drug-like chemical spaces more comprehensively. It adopts a pretraining mechanism using ZINC dataset, where A-GFN learns from inexpensive yet informative molecular descriptors such as drug-likeness, and presents a goal-conditioned finetuning process to adapt A-GFN for downstream optimization tasks.

**Claims And Evidence:**

The authors provide detailed experimental results that support the claim of proposed A-GFN framework in tasks including property optimization, property targeting, and property-constrained optimization.

**Essential References Not Discussed:**

n/a

**Experimental Designs Or Analyses:**

The authors provide detailed descriptions of their experiments on the ZINC dataset pretraining and 3 downstream optimization tasks, comparing their approach to fragment-based GFlowNets, REINVENT, and GraphGA. The ablation studies and sensitivity analyses provide valuable insights into the impact of different design choices

**Methods And Evaluation Criteria:**

The methods and evaluation criteria are well-aligned with the molecular generation task.

**Other Comments Or Suggestions:**

Please refer to weakness

**Other Strengths And Weaknesses:**

Strengths:
1. The paper introduces Atomic GFlowNets (A-GFN), a novel generative model that leverages individual atoms as building blocks, allowing for a more comprehensive exploration of drug-like chemical spaces compared to traditional fragment-based methods.
2. The authors provide a thorough evaluation of A-GFN across various tasks, including property optimization, property targeting, and property-constrained optimization, showing superior performance compared to other methods.

Weaknesses:
1. Figures (e.g., Figure 3,5,6,7,8,9, 10) are blurry or inconsistently scaled, hindering interpretation of molecular structures and training dynamics.
2. Post-hoc validity corrections (e.g., RDKit filtering) during finetuning muddy the contribution of atomic-level modeling. This undermines claims about the standalone effectiveness of the pretrained policy.
For instance, the fragment-based GFlowNet baseline might not explicitly enforce validity via RDKit during generation, while A-GFN inherently guarantees validity by design. This creates an unfair advantage for A-GFN.
3. The paper does not provide computational costs comparisons between A-GFN and other methods (fragment-based GFlowNets, REINVENT), preventing a clear assessment of performance-to-resource tradeoffs. Given the substantial pretraining investment (4 A100 GPUs for 12-18 days), explicit efficiency comparisons would help understand if the atomic-level modeling advantages justify these increased computational demands.
4. The paper constructs an extensive framework that somewhat obscures its core contributions. The structure needs to better emphasize how the atomic-level design specifically enhances pretraining capabilities, rather than focusing excessively on comparisons with non-pretrained methods. The introduction of more data through pretraining potentially creates unfair comparisons (For instance, pretraining data from ZINC may partially overlap with evaluation tasks like logP optimization), leading to somewhat trivial conclusions with non-pretraining methods. Additionally, the pretraining comparisons are affected by the design concerns mentioned in 2. regarding validity enforcement. Based on these observations, I believe this work requires further revision to better highlight and clarify its actual contributions.
5. The paper seems not provide detailed descriptions of the implementation of the baseline methods. The baselines, such as fragment-based GFlowNets, MolDQN, Reinvent, and others, are mentioned in the context of comparison, but the paper does not delve into the specifics of how these baseline methods were implemented or fine-tuned. This makes the experimental results difficult to interpret, as differences in implementation details could significantly impact performance comparisons.

**Questions For Authors:**

Please refer to weakness

**Relation To Broader Scientific Literature:**

This paper extends fragment-based GFlowNets to atomic action spaces. Its pretraining aligns with unsupervised RL pretraining but adapts to molecular design via hybrid online-offline learning.

**Theoretical Claims:**

The paper applies existing GFlowNet theory.

---

> ### Author Rebuttal · Authors · 2025-03-30
>
> We thank the reviewer for their constructive feedback. Our response and proposed revisions for the concerns raised by the reviewer are as follows
> # 1
> We will ensure that figure sizes remain consistent throughout the appendix, particularly improving the font readability of Figure 3. Thank you for pointing this out. We have separated the molecules into separate high-resolution images for better legibility:  https://imgur.com/4pAWr4Q; https://imgur.com/a/jGxFnhY; https://imgur.com/a/uroAmCe; https://imgur.com/a/0biixYL. We’ll update the figure in the revised version of the paper. These molecules were sampled from the same pretrained model as before.
> # 2
> Our approach does not involve post-hoc RDKit filtering or penalization. Instead, A-GFN ensures atomic valency correctness at every generation step, preventing invalid states.
> This is crucial for atom-based GFlowNets, which operate in a vast combinatorial space, unlike fragment-based GFN, which is limited to predefined fragments. Without valency constraints, the model would frequently generate infeasible structures, making exploration intractable.
> Molecular validity is just one of many evaluation criteria. The key metric #Modes measures how many diverse molecules satisfy all four pretraining conditionals (TPSA, Num Rings, QED, SAS). A-GFN significantly outperforms fragment-based GFN in this regard, demonstrating superior exploration—not merely benefiting from valency constraints.
> Thus, valency enforcement is an inherent design choice, not a post-hoc filtering step, and does not confer an artificial advantage.
> # 3
> All baselines (Fragment-GFN, REINVENT, MolDQN, GraphMCTS) were fine-tuned under identical compute constraints: 24 hours on a single V100 GPU, ensuring a fair comparison.
> A-GFN pretraining lasted \~12 days on 4 A100 GPUs (250K steps), but a 100K-step checkpoint (\~5 days) was sufficient for fine-tuning, as reward plateaued (see https://imgur.com/a/hLITsk5 ; to be included in the revised paper).
> While pretraining incurs a one-time cost, it benefits multiple downstream tasks. Once pretrained, A-GFN fine-tuning takes just 24 hours on a V100, making it comparable to molecular optimization baselines. This aligns with established pretraining and finetuning practices seen in ChemBERTa and MoLFormer.
>
> # 4.1
> We would like to clarify that our primary contribution is not that atomic-level design enhances pretraining, but rather the other way, i.e., pretraining is essential for A-GFN to function effectively. Without pretraining, A-GFN fails to generate viable molecules. Our experiments confirm that training A-GFN from scratch (Task train AGFN) consistently fails to satisfy task constraints.
> # 4.2
> Using ZINC for pretraining could introduce overlap with evaluation tasks like logP optimization, but it is common in pretraining large models  to have some solutions for the downstream task to be contained in large scale pretraining data as a broadly representative pretraining data distribution is desired.  Zero-shot OOD molecule generation is non-trivial, even for experts.
> To ensure benchmarking remains challenging, we chose property thresholds so that <25% of ZINC molecules meet the optimization criteria. Additionally, our Novelty metric measures unique molecules not present in ZINC, confirming that solutions stem from exploration rather than memorization.
> # 4.3
> Some baselines (Fragment-GFN, REINVENT) were pretrained using their respective methods, while others (MolDQN, GraphMCTS) were trained from scratch, per the PMO benchmark (https://github.com/wenhao-gao/mol_opt/tree/main) . We used publicly available pretrained models where applicable, ensuring fair comparisons. A-GFN’s superior performance is due to better exploration, not an unfair pretraining advantage.
>
> # 5
> We acknowledge that our paper does not include detailed descriptions of the baseline implementations, and we appreciate you bringing this to our attention. All baseline methods used in our work were directly adopted from the publicly available GitHub repository: https://github.com/wenhao-gao/mol_opt/tree/main. This repository is associated with the preprint https://arxiv.org/pdf/2206.12411, which includes some relevant implementation details, such as whether a method involves pretraining (check Table 7). However, specific aspects such as pretraining compute or further fine-tuning configurations are not comprehensively documented there either. We will add these details in the revised version of the paper for better transparency and reproducibility.

---

> > ### Comment · Reviewer_uZjw · 2025-04-07
> >
> > I appreciate the authors' responses to my concerns. The clarifications regarding figure quality, validity enforcement mechanisms, and computational constraints are helpful.
> >
> > However, I still have reservations about the presentation of metrics and baseline comparisons. While the paper contains extensive tables and metrics, many specialized terms and evaluation criteria remain insufficiently explained, making it difficult for readers not deeply familiar with the field to assess the significance of the results. For instance, the case study on page 7 featuring five targets shows impressive performance, but without adequate explanation of these targets' significance or the meaning of the specific metrics used, the impact is diminished. In the revised version, I would strongly encourage the authors to: (1) Provide more thorough explanations of specialized metrics and their significance; (2) Offer more context for the case studies and their relevance to the field; (3) Include a more accessible discussion of results for readers who may not be intimately familiar with all molecular optimization metrics.
> >
> > With these improvements, the paper would be significantly strengthened and more accessible to a broader audience, especially for the readers from the ML community. Given the thoroughness of the experimental work and the novelty of the approach, I am adjusting my recommendation slightly upward.

---

### Official Review · Reviewer_sFAt · 2025-03-10

**Overall Recommendation:** 4

**Summary:**

This paper proposes a training strategy to improve GFlowNet-based molecular generation. First, it uses atom-based policy rather than fragment-based policy to enable access to a larger chemical space. Second, this work proposes using expert trajectories constructed from ZINC to pretrain the network, which improves drug-likeness and sampling efficiency. Third, this work also proposes pre-training the policy network using inexpensive rewards (including QED, NumRings, TPSA, SAS, which can be computed instantly with RDKit). Such pretraining improves the sampling performance for more complex oracle scores.

**Claims And Evidence:**

- The first claim of this work is the atom-based model enables exploration of larger chemical space. Table 2 supports this claim, with more modes discovered by A-GFN and higher diversity compared to fragment-based models.
- The second claim of this work is the unsupervised pretraining strategy "*enables broader exploration of chemical space while maintaining diversity and novelty relative to existing drug-like molecule datasets*". This claim has been supported by results presented in Table 3, Table 4, etc, which shows that pretraining the model using trajectories derived from ZINC and inexpensive property conditioning can improve sampling.
- The last claim is the fine-tuning strategy and its effectiveness in multiple applications, which has also been well-supported by experimental results presented in Section 6.4 and Section 7.
- Overall, the main claims of this work have been well-supported by experimental evidence.

**Essential References Not Discussed:**

N/A

**Experimental Designs Or Analyses:**

No remarks.

**Methods And Evaluation Criteria:**

- The proposed method is well-motivated by insights into molecular generation and it is a notable contribution to the area of GFlowNet's application in molecular design. Notably, it makes sense that "*these rewards are computationally cheap to evaluate and serve as proxies for more complex properties*", as complex properties have the foundation of physico-chemical properties which are usually inexpensive to compute.
- The overall framework is well-motivated, but the components are mostly based on existing techniques.
- The evaluation is comprehensive and supports the main claims of this work. Justification of each component and demonstration of application are both provided.

**Other Comments Or Suggestions:**

N/A

**Other Strengths And Weaknesses:**

N/A

**Questions For Authors:**

N/A

**Relation To Broader Scientific Literature:**

This work is a contribution to both GFlowNet application and molecular generation.

**Theoretical Claims:**

N/A

---

> ### Author Rebuttal · Authors · 2025-03-30
>
> We sincerely appreciate your thoughtful and constructive review of our paper. We are pleased to see that you find our work well-motivated, comprehensive in evaluation, and a notable contribution to the application of GFlowNets in molecular design. Additionally, we are grateful for your recognition that our claims are well-supported by experimental evidence, with no significant concerns raised regarding our methodology, theoretical foundation, or experimental design.  Based on the feedback from other reviewers, we have further improved our paper to enhance clarity, strengthen key arguments,
>
> Given that no major weaknesses have been identified in your review and that our contributions are acknowledged as valuable to the field, we kindly ask you to reconsider and raise your current score. Your support would help ensure that this contribution reaches the broader community and advances research at the intersection of GFlowNets and molecular generation.

---

### Official Review · Reviewer_tSwN · 2025-03-14

**Overall Recommendation:** 4

**Summary:**

This paper introduces Atomic GFlowNets (or A-GFNs), an atom-based generative framework for molecular design based on GFlowNets, proposing a more general-purpose exploration of the chemical space. The authors propose pre-training A-GFNs on inexpensive molecular properties that act as rewards for training the underlying GFlowNet policy, and then fine-tune for use on a variety of downstream tasks. The authors show the effectiveness of the proposed method on a variety of drug design tasks.

**Claims And Evidence:**

Claims are well-supported by clear and convincing experiments.

**Essential References Not Discussed:**

None to the best of my knowledge.

**Experimental Designs Or Analyses:**

Yes, the experiment design is sound and extensive overall.

**Methods And Evaluation Criteria:**

The atom-based action space makes sense for exploring novel scaffolds, albeit potentially increasing state space complexity. The hybrid online-offline approach and RTB-finetuning are well-motivated and empirically validated in experiments. The use of inexpensive rewards as proxies for general properties is pragmatic and promising for scaling. The evaluation and baselines are comprehensive and properly demonstrate the effectiveness of the proposed method.

**Other Comments Or Suggestions:**

None to be particularly mentioned here.

**Other Strengths And Weaknesses:**

None to be particularly mentioned here.

**Questions For Authors:**

- Why does TB sometimes outperform RTB in single-objective tasks (Table 3)? Is it due to over-regularization in RTB?
- How sensitive is the method to the choice of inexpensive reward descriptors?
- How would a fragment-structured GFN compare to A-GFN?

**Relation To Broader Scientific Literature:**

The main contributions are very relevant for designing general-purpose molecular generation models that can be fine-tuned for a variety of useful downstream tasks.

**Theoretical Claims:**

None to be discussed.

---

> ### Author Rebuttal · Authors · 2025-03-30
>
> 1. Why does TB sometimes outperform RTB in single-objective tasks (Table 3)? Is it due to over-regularization in RTB?
>
> Thank you for raising this important question. The observed performance difference stems from fundamental differences in how TB and RTB balance optimization objectives:
> Yes, RTB's design introduces over-regularization in single-objective tasks due to its explicit anchoring to the pretrained prior. While this prior (e.g., a chemical validity model) helps maintain desirable properties in multi-objective settings, it creates conflicting constraints when optimizing for a single objective (which does not capture these desirable properties). We made initial efforts to address this trade-off in our current draft (lines 297-303), where we discuss how RTB’s prior anchoring leads to its weaker performance when compared to TB. We would like to point out that this effect would diminish when the prior already overlaps with high-reward solutions, and/or if tasks require balancing multiple constraints (e.g., both chemistry rules and potency). We appreciate this opportunity to clarify and will emphasize this trade-off more explicitly in our revision.
>
> 2. How sensitive is the method to the choice of inexpensive reward descriptors?
>
> This is an excellent question, and we appreciate the reviewer's interest in understanding the sensitivity of our method to the choice of inexpensive reward descriptors. The selection of pretraining properties plays a crucial role in shaping the learned policy, and our choices were guided by well-established principles in drug discovery. Specifically, we selected QED (quantitative estimate of drug-likeness), SAS (synthetic accessibility score), TPSA (topological polar surface area), and the number of rings, as these properties are widely used in molecular generation and optimization tasks due to their strong correlation with drug-likeness and pharmacokinetic properties [1].
>
> Empirically, we found that these four descriptors provided a well-balanced pretraining signal that improved sampling efficiency while preserving chemical diversity. Importantly, we observed that adding additional constraints led to premature convergence and mode collapse, as the optimization problem became overly restrictive, significantly reducing the number of valid and diverse molecules that could be generated. This aligns with findings in other large-scale generative modeling approaches, where excessive constraints during pretraining can limit the exploration capacity of the model and degrade its generalization ability in downstream fine-tuning.
>
> 3. How would a fragment-structured GFN compare to A-GFN?
>
> We thank the reviewer for this question. The finetuned A-GFN generally outperforms fragment based GFN across all tasks considered in terms of diversity of molecules as well as the success percentage, which quantifies the ratio of molecules simultaneously satisfying all the enforced objectives. We have detailed these results in Table 12- 20 in Appendix F.
>
> [1] Wellnitz, James, et al. "STOPLIGHT: a hit scoring calculator." Journal of Chemical Information and Modeling 64.11 (2024): 4387-4391.

---

### Decision · Program_Chairs · 2025-05-01

**Decision:**

Accept (poster)

**Comment:**

This paper proppose  Atomic GFlowNets (A-GFNs), a foundational generative model leveraging individual atoms as building blocks to explore drug-like chemical space.  The authors pre-train A-GFN on inexpensive molecular properties as rewards, and then further fine-tune on  downstream tasks. The authors show the effectiveness of the proposed method on a variety of drug design tasks.

Strengths of the paper:
- The proposed method is well-motivated by insights into molecular generation.
- The rewards are computationally efficient.
- The paper provides a thorough evaluation of A-GFN across various tasks, including property optimization, property targeting, and property-constrained optimization, showing superior performance compared to other methods.

Weaknesses of the paper:
- The overall framework's components are based on existing techniques.
- Post-hoc validity corrections (e.g., RDKit filtering) during finetuning undermine the claimed effectiveness of the pretrained policy.